# The genetic mechanism of B chromosome drive in rye illuminated by chromosome-scale assembly

Jianyong Chen[1] ✉, Jan Bartoš [2], Anastassia Boudichevskaia [1,3], Anna Voigt[1], Mark Timothy Rabanus-Wallace [1,4], Steven Dreissig[5], Zuzana Tulpová [2], Hana Šimková [2], Jiří Macas [6], Gihwan Kim [1], Jonas Buhl[7,8], Katharina Bürstenbinder[7,8], Frank R. Blattner [1], Jörg Fuchs [1], Thomas Schmutzer [5], Axel Himmelbach[1], Veit Schubert [1] & Andreas Houben [1] ✉

The genomes of many plants, animals, and fungi frequently comprise dispensable B chromosomes that rely upon various chromosomal drive mechanisms to counteract the tendency of non-essential genetic elements to be purged over time. The B chromosome of rye – a model system for nearly a century – undergoes targeted nondisjunction during first pollen mitosis, favouring segregation into the generative nucleus, thus increasing their numbers over generations. However, the genetic mechanisms underlying this process are poorly understood. Here, using a newly-assembled, ~430 Mb-long rye B chromosome pseudomolecule, we identify five candidate genes whose role as trans-acting moderators of the chromosomal drive is supported by karyotyping, chromosome drive analysis and comparative RNA-seq. Among them, we identify *DCR28*, coding a microtubule-associated protein related to cell division, and detect this gene also in the B chromosome of *Aegilops speltoides*. The *DCR28* gene family is neo-functionalised and serially-duplicated with 15 B chromosome-located copies that are uniquely highly expressed in the first pollen mitosis of rye.

Supernumerary 'B' chromosomes, unlike 'A' (standard) chromosomes, are not required for the normal growth and development of organisms[1,2]. As of 2023, B chromosomes have been discovered in 2951 species from all eukaryotic phyla[3]. Most B chromosomes confer no detectable selective consequences at low numbers, but increased numbers can result in phenotypic aberrations and reduced fertility[4]. B chromosomes are considered parasitic because they have mechanisms

that allow them to use the cellular machinery of the host to achieve their own transmission, even if they are neutral or detrimental to the host. To avoid elimination, many B chromosomes exhibit a type of non-Mendelian inheritance termed *chromosome drive*, whereby segregation during cell divisions before, during, or after meiosis is biased in favor of increasing the number of B chromosomes in germinal cells[5]. The long-term evolution of B chromosomes is likely the outcome of

[1]Leibniz Institute of Plant Genetics and Crop Plant Research (IPK) Gatersleben, Seeland, Germany. [2]Institute of Experimental Botany of the Czech Academy of Sciences, Centre of Plant Structural and Functional Genomics, Olomouc, Czech Republic. [3]KWS SAAT SE & Co. KGaA, Einbeck, Germany. [4]School of Agriculture, Forestry, and Ecosystem Science (SAFES), The University of Melbourne, Parkville, VIC, Australia. [5]Institute of Agricultural and Nutritional Sciences, Martin Luther University Halle-Wittenberg, Halle (Saale), Germany. [6]Biology Centre, Czech Academy of Sciences, Ceske Budejovice, Czech Republic. [7]Department of Molecular Signal Processing, Leibniz Institute of Plant Biochemistry, Halle (Saale), Germany. [8]Institute of Biology, Department of Plant Cell Biology, Philipps University Marburg, Marburg, Germany. ✉e-mail: chenj@ipk-gatersleben.de; houben@ipk-gatersleben.de

selection on the host genome to eliminate B chromosomes or suppress their effects and on the ability of the B chromosomes to escape through the generation of new variants[6]. This highly dynamic mode of evolution is consistent with the extreme variation in B chromosome numbers between species, individuals, and tissues within individuals.

Drive mechanisms in B chromosome systems have been identified in many species and contexts using a range of technologies involving classical genetics and cytogenetics[7,8]. But despite being an ideal test case for the study of drive mechanisms, B chromosome research has only slowly been able to capitalize on the data explosion in sequencing. B chromosomes are structurally highly complex, repetitive, and multitudinous[9], all of which make them demanding for pseudomolecule-level genome assembly, especially before recent developments in the area of long-read sequencing. As such, gene-level insight into the specific mechanisms that control genetic drive is severely limited, although a recent effort by Blavet et al. succeeded in producing a draft assembly of a maize B chromosome that narrowed down the cis and trans sites involved with its drive mechanism[10]. The only known link between a B chromosome-encoded gene and chromosomal drive is still the gene *haploidizer* in the parasitic wasp *Nasonia vitripennis*[11].

The ~560 Mb large B chromosome of rye arose ~1.1–1.3 Mya, coinciding with the origin of the genus *Secale*[12,13]. The rye B is rich in pseudogene-like fragments originating from different rye A chromosomes as well as mitochondria- and chloroplast-derived DNA[13,14]. It also encodes functional genes, such as Argonaute 4, with in vitro RNA slicer activity, which may contribute to RNA-directed DNA methylation of the host genome[15]. Comparative RNA-seq and proteome analysis demonstrated that the rye B chromosome affects the transcriptome and proteome of the host genome[16,17].

At the cell level, the rye B chromosome's drive mechanism takes advantage of asymmetrical spindle formation during the first pollen mitosis (PMI). During this stage, the nucleus of the haploid gamete divides into a heritable generative nucleus and a non-heritable vegetative nucleus. At PMI, rye B chromatids exhibit nondisjunction, remain lagging at anaphase, and finally preferentially enter the closer generative nucleus[18–20] (Fig. 1a). In the second pollen mitosis (PMII), the generative nucleus divides to produce two sperm nuclei, each with an unreduced number of Bs. The nondisjunction of B sister chromatids is probably caused by extended pericentromeric cohesion, which is under the control of the B-specific drive control region (DCR)[20]. Analysis involving deficient B chromosomes revealed that the DCR is located at the end of the long B arm; B chromosome variants lacking a DCR undergo normal disjunction at the first pollen anaphase[21–24] (Fig. 1b). Drive control acts in trans since long arm deficient B chromosomes (defBs) recover drive in the presence of a standard B chromosome[21,24]. The DCR undergoes DNA replication last and is enriched in B-specific satellite repeats and the euchromatin-specific

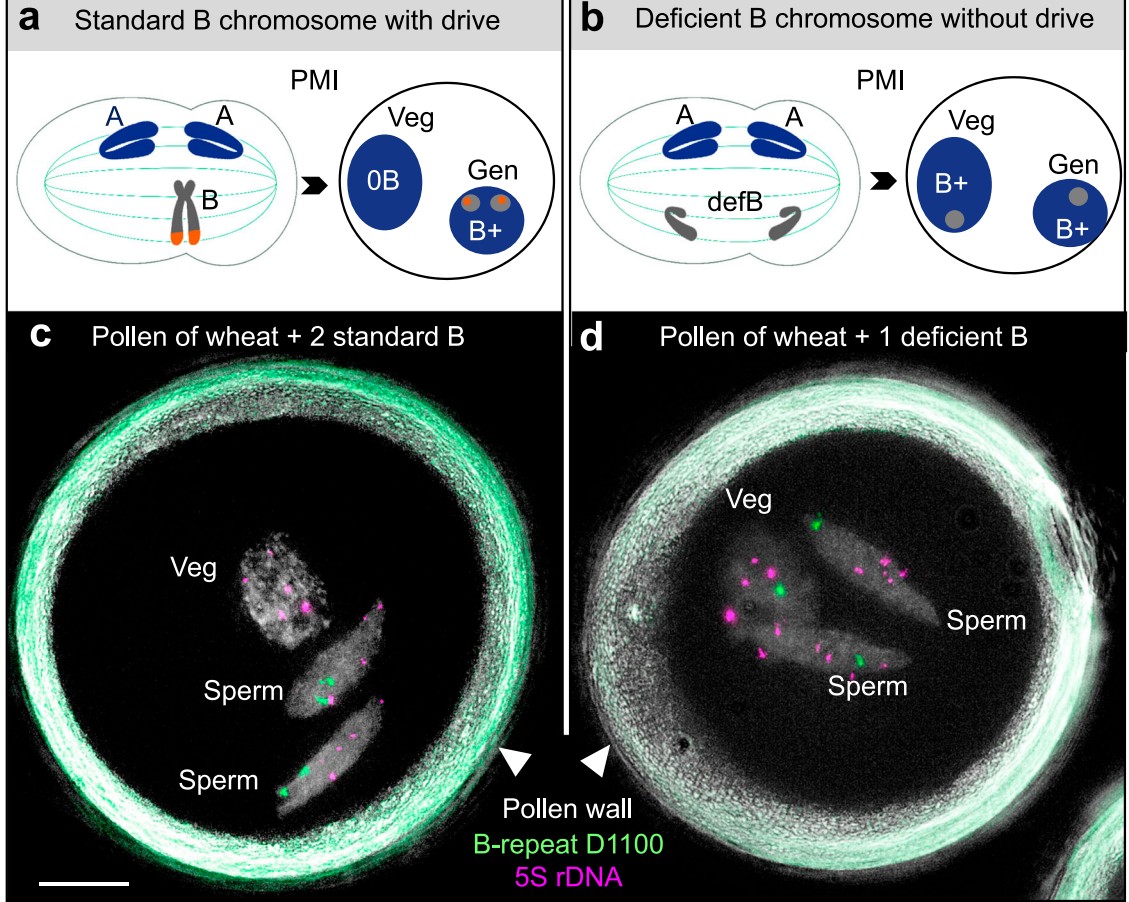

**Fig. 1 | The drive of the rye B chromosome during first pollen mitosis.** The schema depicts the contrasting segregation behavior of (**a**) a drive-positive, standard rye B chromosome undergoing nondisjunction and accumulation in the generative nucleus and **b** a drive-negative, drive control region-deficient rye B chromosome (defB) during the first pollen mitosis (PMI). Standard A chromosomes are shown in blue, B chromosomes in gray, and the nondisjunction control region in red. Veg vegetative nucleus, Gen generative nucleus. Pollen-FISH combined with super-resolution microscopy (3D-SIM) was used to determine the distribution of Bs in mature wheat pollen with two rye drive-positive Bs (**c**) and with a drive-negative, deficient B^k–1 chromosome variant (**d**). Repeat D1100 (green) indicates the B chromosome and 5S rDNA (magenta) is an A chromosome-specific marker. Supplementary Movie 1 shows (**c**) in detail. The experiment was independently repeated twice with similar results. Bar = 10 μm.

**Table 1 | Quantitative analysis of the rye B chromosome drive**

| B variant[a] | Total[b] | No B-specific signal[c] | Only in VN[d] | Only in SN[e] | In all nuclei[f] | Drive? | Drive frequency[g] |
|---|---|---|---|---|---|---|---|
| 1 B[k]–1 | 195 | 103 | 0 | 0 | 92 | No | 0% |
| 2 B[k]–2 | 77 | 15 | 3 | 55 | 4 | Yes | 88.7% |
| 2 B[k]–3 | 101 | 20 | 0 | 0 | 81 | No | 0% |
| 2 B[s] | 112 | 11 | 6 | 91 | 4 | Yes | 90.1% |
| 2 B[k] | 122 | 21 | 6 | 92 | 3 | Yes | 91.1% |

Wheat lines with different B variants were analyzed by pollen-FISH using the B-specific probe D1100. Refer to Fig. 1 for representative examples of pollen with and without B chromosome drive.

*VN* vegetative nucleus, *SN* sperm nuclei.

[a]Wheat possessing different rye B chromosome variants and B numbers.

[b]Total number of analyzed pollen.

[c]Number of pollen without B-specific signals. Because of meiotic/post-meiotic loss of Bs, not all pollen showed B-specific FISH signals.

[d]After B chromosome nondisjunction, B chromatids enter only the vegetative nucleus (PMI).

[e]After B chromosome nondisjunction, B chromatids enter only the generated nucleus (PMI) and subsequently only sperm nuclei (PMII).

[f]B chromatids separate normally (disjoin) during PMI and consequently distribute evenly in all nuclei.

[g]Frequency of pollen with nondisjunction toward generative nucleus among pollen nuclei with B-signals.

posttranslational histone mark H3K4me3[25,26]. The DCR-specific repeat families E3900 and D1100 are transcriptionally active and lack any open reading frames[25]. In rye, the B chromosome drive is highly efficient, with 88%–96% of segregation events favoring the generative nucleus across different rye genotypic backgrounds[27,28]. The rye B chromosome's drive behavior is unchanged when it was introduced as an additional chromosome into hexaploid wheat[21,29]. Thus, the rye B chromosome itself controls the process of nondisjunction[18,30].

In this work, to identify the trans-acting drive-controlling factor(s) on the rye B chromosome, we first narrow down the size of the DCR by applying a set of B-specific fluorescence in situ hybridization (FISH) probes to a diverse panel of drive-positive and drive-negative B variants. Next, we assemble the rye B into a single ~430 Mb-long pseudomolecule from PacBio HiFi long reads and Nanopore ultra-long reads of a wheat-rye B chromosome addition line, supplemented by Hi-C, optical, and repeat mapping. Finally, a tissue-specific, comparative RNA-seq analysis using B chromosome variants, both with and without the ability to drive, reveal five drive-controlling candidate genes embedded in the B repeat-enriched DCR. The genes identified by the transcription analysis are then explored regarding function and evolution using a variety of methods, ultimately suggesting that the rye B gene *DCR28*, derived from a functional A chromosome ancestor, evolved on the B chromosome into playing a key role in controlling drive by modifying the normal functioning of microtubules during first pollen mitosis.

## Results

### Identification of the chromosome region that regulates the B drive

Rye B chromosome variants with and without drive behavior (drive: B[s], B[k], and B[k]–2; non-drive: B[s]–8 and B[k]–1) were established in the background of wheat[21]. B[s] and B[k] represent standard B chromosomes identified in different rye genotypes. The variants B[k]–1 and B[k]–2 are rye B/wheat A translocation chromosomes with missing subtelomeric regions of the long B arm[21]. A previously-undocumented deficient B variant named B[k]–3 was identified when we screened the progeny of wheat with B[k]–2.

Drive status was assayed based on relative accumulation levels of B chromosomes in the pollen sperm vs. vegetative nuclei. We performed FISH on mature pollen using probes targeting the

B-specific D1100 satellite repeat and 5S ribosomal DNA (rDNA), as an A-specific control probe. Drive is evident if sperm nuclei show D1100 signals, but not the vegetative nucleus (Fig. 1c and Supplementary Movie 1). On the other hand, no B chromosome drive occurs if both types of nuclei exhibit D1100 signals (Fig. 1d). Pollen-FISH revealed that 89–91% of pollen showed sperm-specific accumulation of the B variants B[s], B[k], and B[k]–2, while there was no sperm-specific accumulation of B[k]–1 and B[k]–3 (Table 1). Thus, B[s], B[k], and B[k]–2 can efficiently drive in wheat, comparable to the frequency of standard B chromosomes in the background of different rye genotypes[27,28].

To cytogenetically identify the DCR, we performed FISH to compare the composition of the long arm for all six B variants (B[s], B[k], B[s]–8, B[k]–1, B[k]–2, B[k]–3) using probes targeting the rye genome-specific repeat Revolver, and the B-specific repeats D1100, Sc9c130, E3900, and Sc26c38 (Fig. 2 and Supplementary Fig. 1). Sc26c38, E3900, and Sc9c130 positive chromosome regions are adjacent to each other in linear order, and E3900 and D1100 repeats intermingle (Fig. 2b)[26]. FISH revealed that the rye B/wheat A translocation chromosomes B[k]–1 and B[k]–2 possess translocation breakpoints in the terminal region of the long B arm. All drive-positive B variants (B[s], B[k], B[k]–2) showed D1100, Sc26c38, and E3900 signals. Sc26c38 is found also in the drive-negative B[k]–1 and B[k]–3 (Fig. 2 and Supplementary Fig. 2), but E3900 signals were absent in all drive-negative variants (B[s]–8, B[k]–1, B[k]–3). Drive-negative B[s]–8 lost the D1100 signals completely, and drive-negative B[k]–1 and B[k]–3 showed a truncated D1100 region. In addition, a small gap almost in the middle of the D1100 region is present in drive-negative B[k]–3 but not in B[k]–1 (Fig. 2a). We conclude, therefore, that B[k]–3 is the largest B variant among the drive-negative B variants. The B-specific subtelomeric repeat Sc9c130 is absent in all drive-negative variants as well as in B[k]–2 (Supplementary Fig. 3), indicating that the distal region of the long B arm is not required for drive. Thus, the DCR corresponds to the E3900, Sc26c38, and D1100 repeat-containing chromosome region between the breakpoints of B[k]–3 and B[k]–2 (Fig. 2b).

### Assembly of the rye B chromosome pseudomolecule

B chromosomes are notoriously difficult to sequence and assemble owing to their complex repeat structure and high similarity with the A chromosome sequence of the same species. To avoid confounding by the A chromosomes of rye, we sequenced the rye B chromosome in the background of wheat (cv. Chinese Spring); sequence similarity between the A chromosomes of wheat and the rye B chromosome is significantly lower than between the A and B chromosomes of rye, making it easier to differentiate between wheat A and rye B-derived sequence. A rye B addition line was screened by FISH to select plants possessing six standard rye B[s] copies, in addition to 21 pairs of wheat A chromosomes. Several plants contained an additional truncated rye B (Supplementary Fig. 4).

An initial 15-Gb primary assembly was constructed from highly accurate PacBio HiFi and ultra-long Oxford Nanopore reads using the assembly tool hifiasm and achieved a BUSCO completeness of 96.0% based on 2147 contigs with an N50 length of 43.6 Mb (L50 = 85; Supplementary Table 1). The logic that the multiplicity of B chromosomes per cell outnumbered that of any given wheat chromosome threefold is confirmed by the clear bimodal distribution of mean HiFi read coverage levels per contig and the ~1:3 ratio of wheat A (~14x): rye B (~44x) coverage level (Fig. 3a). Hi-C scaffolding resulted in 21 long scaffolds (>500 Mb) (Supplementary Fig. 5) and their alignments to the reference genome of wheat cv. Chinese Spring (IWGSC RefSeq v2.1 assembly) revealed their 1 to 1 syntenic correspondence to near-complete wheat chromosomes (Supplementary Fig. 6; Supplementary Table 2). Contigs accounting for ~452 Mb were assigned to the rye B based on HiFi read coverage (B - coverage > 23x) as none of the wheat pseudomolecules were found to comprise any of the contigs - coverage > 23x (Fig. 3b; Table 2; Supplementary Data 1).

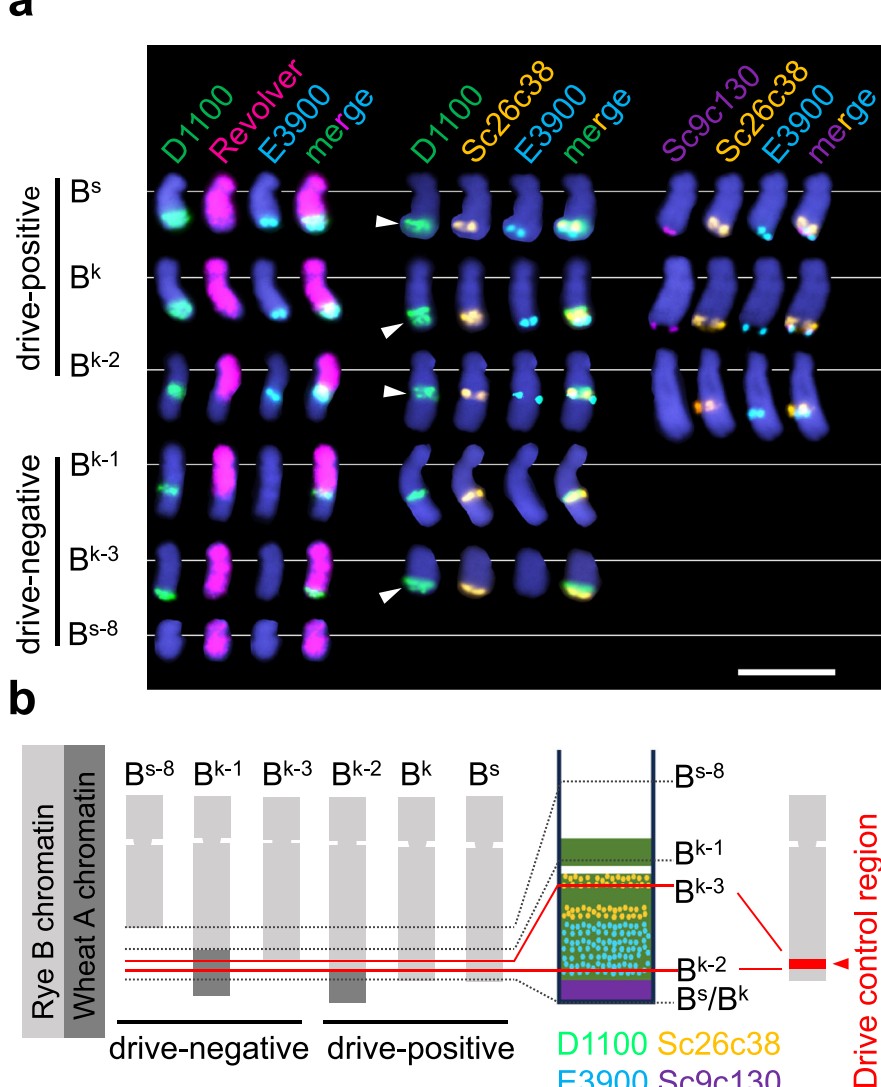

**Fig. 2 | The subtelomeric region of the long arm of the rye B chromosome controls the drive. a** The repeat composition of the terminal region of the long B chromosome arm of the drive-positive B variants B$^s$, B$^k$, B$^k$−2 and the drive-negative B variants B$^s$−8, B$^k$−1, B$^k$−3 were analyzed by FISH using the rye genome-specific repeat Revolver (magenta), and the B-specific repeats D1100 (green), E3900 (sky blue), Sc9c130 (violet), and Sc26c38 (orange) were used as probes to determine the drive control region. Corresponding complete mitotic cells after FISH are shown in Supplementary Figs. 1–3. The experiment was independently repeated twice with similar results. White arrowheads indicate the signal gap in the D1100-positive region. Chromosomes are counterstained with DAPI (blue). Bar = 10 μm. **b** Schemata of the different rye B chromosome variants showing based on (**a**), the distribution of the B-specific repeats D1100 (green), E3900 (sky blue), Sc9c130 (violet), and Sc26c38 (orange). Light gray and diagonal stripes depict rye and wheat chromatin, respectively. The continuous long black lines represent the ends of B$^k$−8, B$^s$, and B$^k$. The blue dotted lines indicate the ends of B$^k$−1, B$^k$−3, and B$^k$−2. On the right, the chromosomal position of the drive control region is shown as a read region of a drive-positive B chromosome.

The B-assigned contigs were arranged into a draft pseudomolecule order using Hi-C and optical map data (Fig. 3b and Supplementary Data 2). This draft order was further manually organized using repetitive sequences as relative positional landmarks[31]. Eight repeat families with known distributions on the rye B[26] were identified on the contigs (Bilby, CL11, Sc55c1, Sc63c34, D1100, Sc26c38, E3900, and Sc9c130 repeats; mitochondrial and chloroplast DNA fragments); Fig. 3c and Supplementary Data 3), and the order adjusted to accord with both the sequence feature distribution and the Hi-C/Optical Map-based draft order. Thirty-eight contigs were thus arranged into a ~430 Mb large pseudomolecule (Fig. 3b,d; Table 2; Supplementary Data 4), representing ~77% of the rye B size as determined by flow cytometry. We confirmed that the missing portion of the assembly is accounted for by the collapsing of long arrays of extremely repetitive sequences[32]. Aligning the B-associated optical-map reads to the B pseudomolecule revealed the expected presence of repetitive regions

with elevated coverage (Supplementary Fig. 7), and the assembled size of E3900, CL11, and Sc26c38 repeats turned out to be only 67%, 84%, and 87% of the total size estimated from short-read sequencing data (Fig. 3d). On the other hand, evaluation of the B pseudomolecule using the Long Terminal Repeat (LTR) Assembly Index (LAI) resulted in an LAI score of 19.19, which is close to gold quality (LAI > 20)[33].

Annotation of the B chromosome genic sequences was performed based upon ~159 Gb of Illumina short-read RNA-seq data collected from seven tissues (root meristems, anthers at both the PMI and PMII pollen mitotic stages, leaves, spikes undergoing meiosis, young shoots, and immature seeds) of wheat with 2B$^s$ (Supplementary Data 5). A total of 1292 transcriptionally active rye B-encoded sequences were predicted, including 799 encoding predicted proteins longer than 100 residues (Table 2 and Supplementary Data 6). This is likely an underestimate of the total gene number as it does not account for lowly expressed or silenced genes. In addition, the

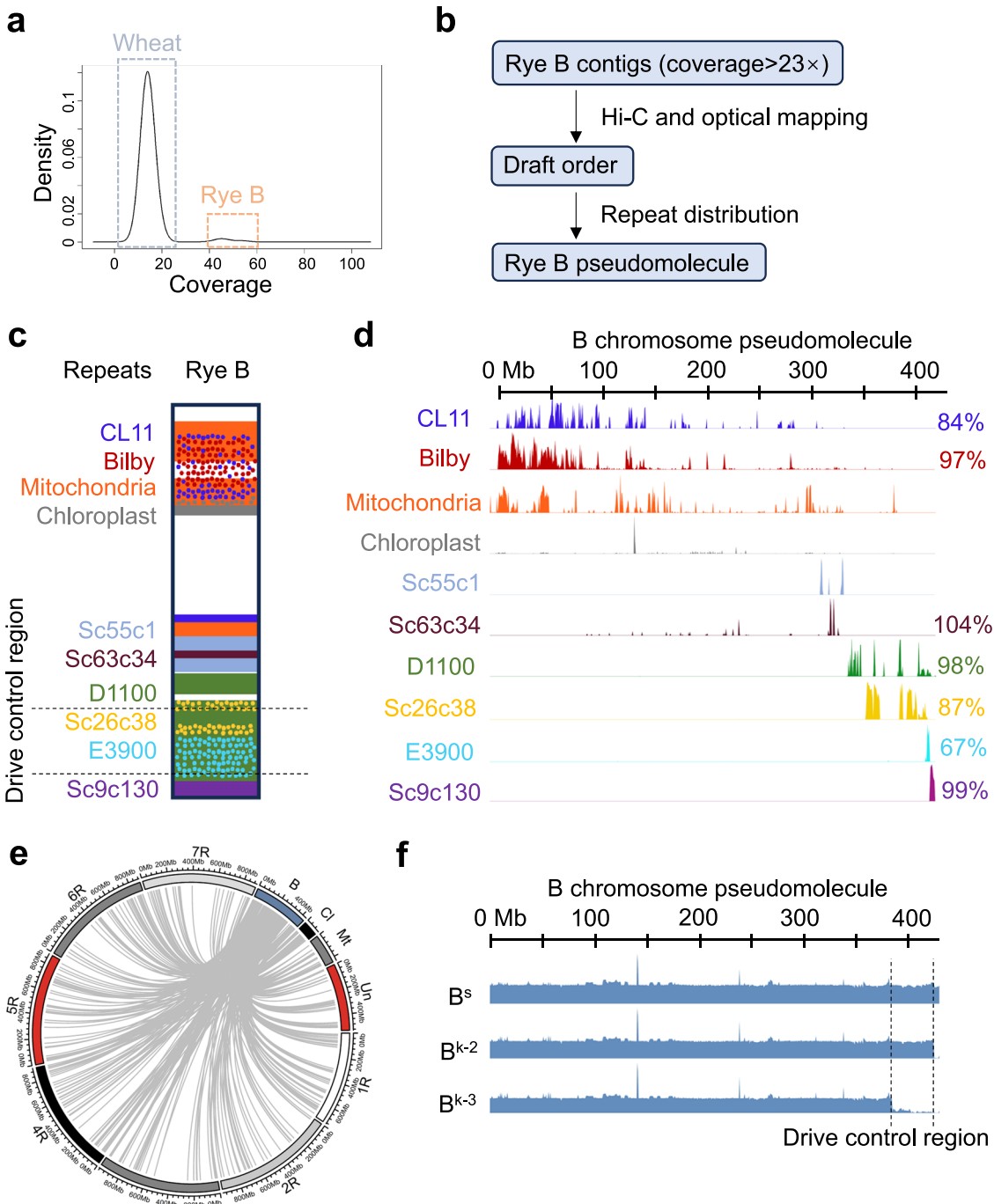

**Fig. 3 | Whole-genome assembly of wheat with the rye B chromosome and arrangement of the rye B-like contigs into a B chromosome pseudomolecule.** **a** Density plot of PacBio HiFi reads coverage across the genome of the wheat-rye B addition line. The higher peak of ~14× represents sequences from the wheat genome. The lower peak of ~44× represents sequences belonging to the rye B chromosome. x-axis: PacBio HiFi reads coverage on the assembled contigs, y-axis: density; **b** the scheme of the scaffolding strategy; **c** a model of repeat distribution on the rye B chromosome; **d** distribution of eight repeats and organellar DNA on the B chromosome pseudomolecule (chrB) in a 1-Mb window, visualized by pyGenomeTracks. Each track represents the abundance of a repeat from minimum to maximum on chrB. x-axis: coordinates on the rye B pseudomolecule. On the right

are the proportions of the assembled size of a repeat/total size estimated from short reads. Organellar DNA was not able to be compared as they also exist in cytoplasm. Sc55c1 was not able to compare as it has similar sequences in the wheat genome. **e** The relationship between the B-located genes and the high-confidence genes on the A chromosome and the organellar genome of rye. B: the rye B chromosome, 1R-7R: rye cv. Lo7 A chromosomes. Un: the unassigned scaffold of the A chromosomes. Cl chloroplast DNA, was enlarged 200 times. Mt mitochondrial DNA was enlarged 100 times. **f** Determination of the drive control region using short-read data from wheat lines with B$^s$, B$^{k-2}$, and B$^{k-3}$. The data mapping was visualized by pyGenomeTracks. The dash lines indicate the location of the rye B drive control region (383,864,014−423,790,502 bp).

annotation of B-encoded genes confirmed the mosaic origin of the rye B chromosome, because the B is composed of sequences derived from all seven rye A chromosomes as well as plastid and mitochondria DNA (Fig. 3e).

## Identification of candidate genes that control the B chromosome drive

The DCR was delimited based on the comparative mapping of sequence reads derived from lines containing drive-positive (B$^{k-2}$, B$^s$)

**Table 2 | The characteristics of the sequence assembly and gene content of the rye B chromosome**

| Rye B contigs | |
|---|---|
| Total number of rye B contigs | 160 |
| Largest contig | 93,027,977 bp |
| Total length of all rye B contigs | 452,394,662 bp |
| N50 | 60,109,561 bp |
| L50 | 3 |
| **Assembled B chromosome pseudomolecule** | |
| Length of the B chromosomal pseudomolecule | 429,632,691 bp |
| Number of scaffolded rye B contigs | 38 |
| Size of the largest scaffolded contig | 93,027,977 bp |
| **Genes encoded by the rye B chromosome (in the background of wheat)** | |
| Number of transcriptionally active genes | 1292 |
| Max, median, min length of transcripts | 44051, 1279, and 201 nt |
| Number of protein-coding genes (ORF > 100 aa) | 799 |
| Max and median length of peptides | 1879 and 199 aa |

The detailed information of the sequence assembly can be found in Supplementary Data 1–4. The detailed information of the rye B genes can be found in Supplementary Data 6.

and drive-negative B chromosomes (B$^k$−3). Approximately 36 Gb (~2× coverage) of short-read genome sequencing data were mapped to the B pseudomolecule and unplaced B contigs, constraining the DCR to a ~40 Mb chromosome segment between 383.9 and 423.8 Mb on the rye B pseudomolecule (Fig. 3f), and 16 unplaced B contigs (totaling 3.7 Mb) (Supplementary Table 3). The location of this segment on our B chromosomal assembly closely corresponded to the results of FISH-mapping. In total, 88 genes (representing 31 single-copy genes and 11 gene families) are located in the DCR of the B pseudomolecule, 66 of which contain open reading frames (Supplementary Data 6). No active genes were annotated on the 16 unplaced contigs.

Expression of genes annotated on their wheat or rye A chromosomes[34,35] was quantified using RNA-seq on a variety of wheats and ryes with varying B karyotypes and drive statuses (Supplementary Data 5). A subtractive stepwise approach was followed. At each step, pair-wise comparisons of B gene expression between a collection of drive-positive and drive-negative samples were performed, and genes were retained only if they showed twofold upregulation in the drive-positive set (adjusted $p$ value < 0.01) (Fig. 4a). Beginning with wheat drive-positive vs. drive-negative comparisons and ending with rye drive-positive vs. drive-negative comparisons, the subset of eligible differentially expressed genes (DEGs) was stepwise reduced from 533 down to 23 genes (Supplementary Data 7 and 8). Sequence-based clustering using CD-HIT indicated these drive-induction candidate genes fall primarily within seven single-copy genes and two gene families with multiple members (Fig. 4a and Supplementary Data 7).

To assess which DCR-encoded gene possesses members in the DCR exclusively, PCR assays were conducted on drive-positive and negative variants, with primers designed to amplify gene-bearing fragments of the DCR. Genomic DNA of wheat possessing either drive-positive (B$^s$, B$^k$, B$^k$−2) or negative (B$^s$−8, B$^k$−1, B$^k$−3) B variants were used as PCR templates. Four genes (DCR398, DCR399, DCR145, and DCR154) appear to be located not only within the DCR but also at other positions on the rye B chromosome (Supplementary Data 7 and Supplementary Fig. 8). Five DCR-located genes (DCR260, DCR169, DCR83, DCR400, and DCR28) produced amplicons only in the presence of drive-positive B variants (Supplementary Data 7 and Supplementary Fig. 8), indicating that their members are exclusively DCR-localized. Since B$^s$−8, B$^k$−1, and B$^k$−3 are drive-negative, we speculate that the trans-acting element(s) known to effect B chromosome drive, are

probably among these five candidates. The activities of these five DCR-exclusive candidate genes were compared across different tissues (Fig. 4b and Supplementary Data 9). DCR28 and DCR83 showed increased activity during PMI, and DCR28 exhibits the highest expression among all candidates. Reverse transcription polymerase chain reaction (RT-PCR) confirmed PMI-specific expression of DCR28 (Supplementary Fig. 9).

### Functional exploration of drive-controlling candidate genes

To associate DCR-located gene sequences with functions based on homology to functionally characterized genes, we used the PANTHER (Protein Analysis Through Evolutionary Relationships) classification system (http://www.pantherdb.org). DCR28 was classified as the microtubule-associated Futsch-like protein family (PTHR34468), an uncharacterized family in plants (Supplementary Data 6). The possible microtubule-associated function of DCR28 raised our interest, as the regulation of microtubule dynamics is key for mitotic spindle assembly and faithful chromosome segregation. DCR400 belongs to the ATP-dependent DNA helicase DDX11-related family (PTHR11472:SF41), which is similarly linked to nondisjunction via its roles in DNA duplex unwinding and sister chromatid cohesion[36]. However, DCR400 not only shows stronger expression in shoot apical meristems (TPM = 2.5) than in the anthers during PMI (TPM = 0.6) but also expresses in root and spike (Fig. 4b). All five transcript variants of DCR169 share similarities with long non-coding RNA-producing B-specific satellite E3900 (Supplementary Fig. 10) and three of them may encode an ~160 aa unknown protein (Supplementary Data 6 and 7). However, the probable gag gene origin of E3900 from a centromere-specific retrotransposon element[37] lends some support that E3900/DCR169 may play a not-yet-defined role in the process of nondisjunction in combination with additional controlling elements. There is no evidence so far that the remaining two candidates are involved in chromosome segregation (Supplementary Data 6): DCR83 encodes a 288-aa protein belonging to the family glycine-rich cell wall structural protein 1.8-like (PTHR31286). DCR260 encodes a 670-aa protein similar to 2-isopropylmalate synthase (PTHR36786).

### DCR28 is also encoded by the *Ae. speltoides* B chromosome

We tested whether the B chromosome of the related species *Aegilops speltoides* (*Ae. speltoides*) also encodes similar first pollen mitosis-active candidates. *Ae. speltoides* B chromosome was selected because of the rye B chromosome-like drive mechanism[38,39]. We searched our five candidates in the de novo assembled transcriptome of anthers undergoing PMI from *Ae. speltoides* with B chromosomes and only identified a DCR28-like gene (called AesDCR28). AesDCR28 is 290-aa long and has 60% (161/265) similar amino acids compared to DCR28 of the rye B (Supplementary Fig. 11 and Supplementary Data 10). Like in rye, AesDCR28 was shorter than the corresponding A chromosome variant (Supplementary Fig. 12). Further, genomic PCR showed the presence of AesDCR28 on all B chromosomes from four different *Ae. speltoides* accessions (Fig. 4c). Furthermore, given the known monophyletic origin of the rye B chromosome[40], and the necessity of a reliable drive mechanism for its persistence over evolutionary time, we reason that the drive mechanism is most likely common across rye lineages and, thus, that a plausible candidate drive gene must have homologs in all +B ryes. To confirm this for DCR28, we analyzed a panel of diverse weedy ryes (*Secale cereale* subsp. *segetale*), which also carry drive-positive B chromosomes[27,40]. The presence of DCR28 was confirmed by genomic PCR in all +B rye genotypes (Fig. 4c). Thus, both B chromosomes of rye and *Ae. speltoides* encode DCR28-like genes.

### The B-specific DCR28 family arose from an A chromosome-located paralog

The B location of DCR28 within the DCR of these weedy ryes and cultivated reference variety was cytogenetically verified by FISH using mitotic

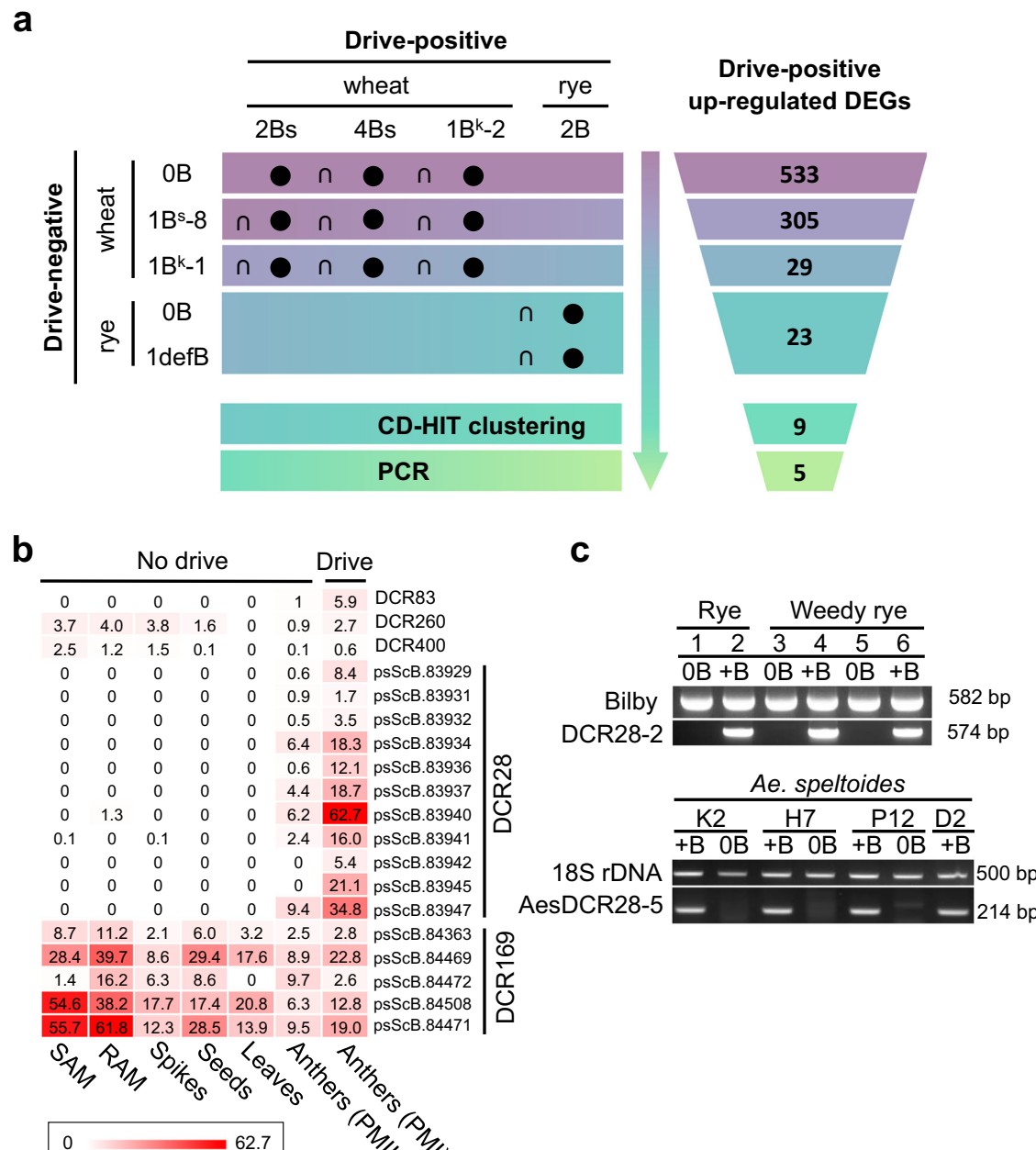

**Fig. 4 | Identification of potential trans-acting factor(s) that control the rye B chromosome drive process by comparative RNA-seq analysis. a** Differential expression analysis between plants with and without B drive was performed in wheat and rye to find commonly drive upregulated genes (fold change > 2, adjusted *p* value < 0.01, method: DESeq2). First, we pair-wise compared the drive-positive genes of wheat carrying the different B numbers and variants (2Bˢ, 4Bˢ, and 1Bᵏ⁻2) with 0B, respectively, and 533 commonly upregulated differentially expressed genes (DEGs) were identified. Next, the genes of wheat with 2Bˢ, 4Bˢ, and 1Bᵏ⁻2 were compared to the genes of the drive-negative wheat + 1Bˢ⁻8, and 305 upregulated DEGs remained. Afterwards, the genes of wheat with 2Bˢ, 4Bˢ, and 1Bᵏ⁻2 were compared with the data of the drive-negative wheat + 1Bᵏ⁻1, and 29 commonly upregulated DEGs were found. Finally, after comparing the data of rye with 2B to the data of 0B rye and rye + 1defB, respectively, a total of 23 commonly upregulated DEGs were found across the 11 comparisons (black circles). The comparisons were one-sided and '∩' indicates intersection among comparisons. Subsequent CD-HIT clustering and PCR further reduced the number of candidates to 9 and 5, respectively. The color code reflects the progressing reduction of candidate genes with each analysis step. **b** expression activity of the five drive candidates (DCRs) in different tissues of wheat + 2Bˢ: shoot apical meristem (SAM), root apical meristem (RAM), young spikes undergoing meiosis, seeds (7 days after pollination), young and old leaves, and anthers undergoing second pollen mitosis (PMII) and the first pollen mitosis (PMI). Expression calculation method: transcripts per million (TPM); **c** B-specific PCR amplicon demonstrates that *DCR28* is B chromosome-specific in rye and *Ae. speltoides*. DNA templates: cultivated rye (1, 2), weedy rye from Afghanistan (3, 4) and Pakistan (5, 6), *Ae. speltoides* K2 (Tartus, Syria), H7 (Katzir, Israel), P12 (Ramat Hanadiv, Israel), D2 (Tartus, Syria). Bilby-specific primers were used as rye DNA control. 18S rDNA primers were used as *Ae. speltoides* DNA control. The experiment was independently repeated twice with similar results. Source data are provided as a Source Data file.

metaphase chromosomes of cultivated and weedy rye with standard B chromosomes and pachytene chromosomes of wheat with Bˢ (Fig. 5a, b and Supplementary Fig. 13). Strong overlap between *DCR28* and Sc26C38/D1100 hybridization signals suggested the presence of clustered *DCR28* copies in the DCR of both species (Fig. 5b and Supplementary Fig. 13). FISH on naturally extended pachytene chromosomes confirmed overlapping *DCR28* and Sc26c38 signals, and demonstrated the distal position of E3900 arrays (Fig. 5b and Supplementary Fig. 13).

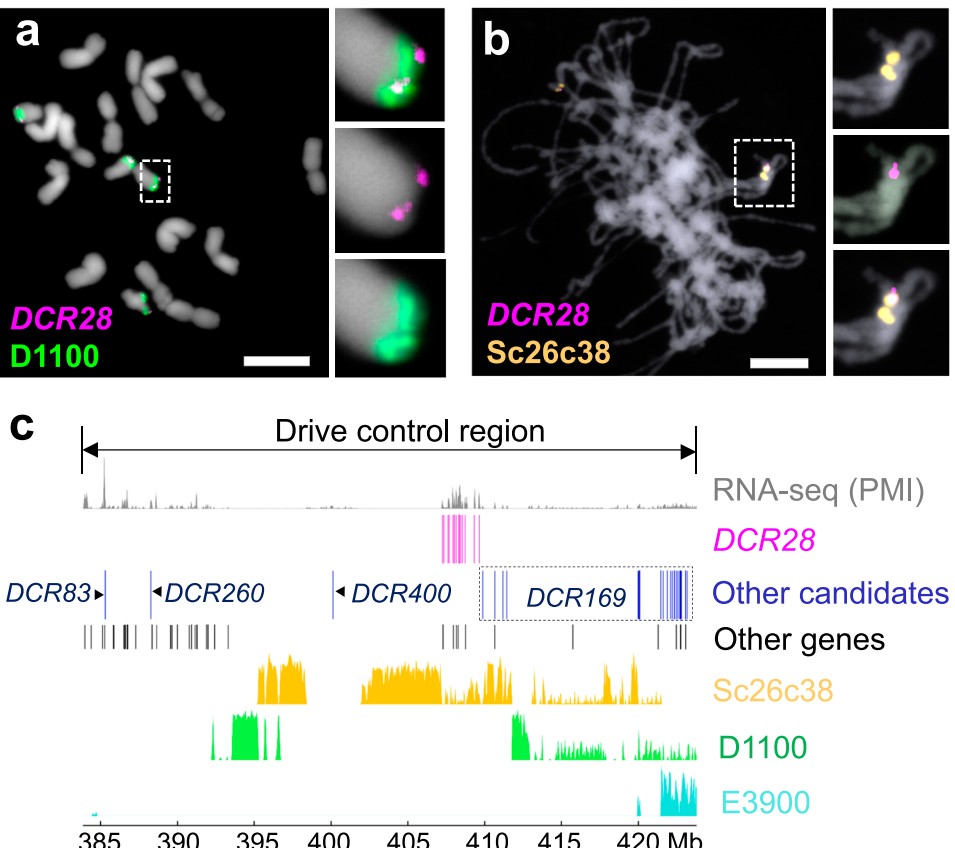

**Fig. 5 | *DCR28* genes intermingle with Sc26c38 repeats within the rye B chromosome drive control region. a** Mitotic metaphase of cultivated rye with four standard B chromosomes showing *DCR28*-specific (magenta) and D1100-specific (green) FISH signals. Bar = 10 μm. **b** Pachytene chromosomes of wheat with three rye Bˢ chromosomes showing *DCR28*-specific (magenta) and Sc26c38-specific (orange) FISH signals. Inlets are showing further enlarged B chromosome regions after FISH. Bar = 10 μm. The FISH experiments were independently repeated twice with similar results. **c** Multiple copies of *DCR28* form a cluster in a 2.3-Mb region. The drive control region (383.9–423.8 Mb) on the chrB was visualized by pyGenomeTracks. The first track (gray) indicates the expression of the RNA-seq data of anthers undergoing the first pollen mitosis (PMI) of wheat with 2Bˢ. The second track (magenta) represents the *DCR28* gene cluster, from 407,247,670 to 409,643,611 bp on chrB. The third track (blue) represents other candidate genes in this region, including three single-copy genes *DCR83*, *DCR260*, *DCR400*, and the *DCR169* gene family (dashed box). The fourth track represents other genes in this region that do not belong to the differentially expressed candidates. The fifth track (orange) represents the distribution of the repeat Sc26c38, and the sixth track (green) represents the distribution of the repeat D1100. The last track (sky blue) represents the distribution of the repeat E3900.

The multiplicity of *DCR28* in the DCR is consistent with diversification as an evolutionary means of competing in an arms race. Local gene clusters are suggestive of duplication events mediated by local microhomologies via, e.g., repeat arrays that lead to strand slippage during replication[41,42]. Alternatively, processes like tandem duplication (e.g., *Slxl1* and *Sly* in mouse[43,44]) or spreading in association with other repetitive DNA or transposable sequences, such as *wtf* spore killers in *Schizosaccharomyces pombe*[45] and *Dox* in *Drosophila*[42] or others played a role.

In the assembled B genotype, members of the *DCR28* family cluster in a 2.3 Mb-long subregion, intermingled with Sc26c38 repeat arrays. Several other genes intermingle with the *DCR28* cluster (Fig. 5c). Three of these genes show close homology to the rye gene SECCE4Rv1G0278580 on chromosome 4R, suggesting, in common with *DCR28*, a copying event followed by local duplication (Supplementary Data 6). Moderate sequence variation exists between *DCR28* members, with predicted proteins varying in length from 122 to 303 residues (Supplementary Fig. 14 and Supplementary Data 6). *DCR400* by contrast, is present in a single copy located ~7 Mb away from *DCR28* (Fig. 5c).

Homology search using BLASTp showed *DCR28* and *DCR400* share recent common ancestry with rye A chromosome-located paralogs; *DCR28* is most closely related to the single-copy gene SECCE7Rv1G047916 (called *DCR28-like rye*) on rye chromosome 7R, though *DCR28* encodes a protein 71 residues shorter than its A chromosome counterpart (Supplementary Fig. 15). *DCR400* shows exceptionally high amino acid similarity to the rye paralog gene SECCE1Rv1G0060580 (called *DCR400-like rye*) of A chromosome 1R (Supplementary Fig. 16) suggesting a more recent copying event or the existence of high functional constrains for B chromosome drive. *DCR28*-like genes and *DCR400*-like genes also possess homologs in all three wheat genomes. *DCR28* shares closest homology with TraesCS4A03G0224100 (*DCR28-like wheat A*), TraesCS4B03G0550000 (*DCR28-like wheat B*), TraesCS4D03G0484200 (*DCR28-like wheat D*), and *DCR400*-like shares closest homology with TraesCS1D03G0983600 (*DCR400-like wheat D*), TraesCS1B03G1206100 (*DCR400-like wheat B*), and TraesCS1A03G1024600 (*DCR400-like wheat A*) (Supplementary Figs. 15 and 16).

Compared with the *DCR28* family, the rye A variant *DCR28*-like expressed at a much lower level during the PMI (Fig. 6a), while *DCR400* and *DCR400*-like rye both express at a very low level (TPM: 2-3) during PMI (Fig. 6a). The expression patterns of the three A-paralogs of *DCR28* in wheat (*DCR28-like wheat A, B, D*) are similar: They showed high expression in young meiotic spikelets and lower expression in other

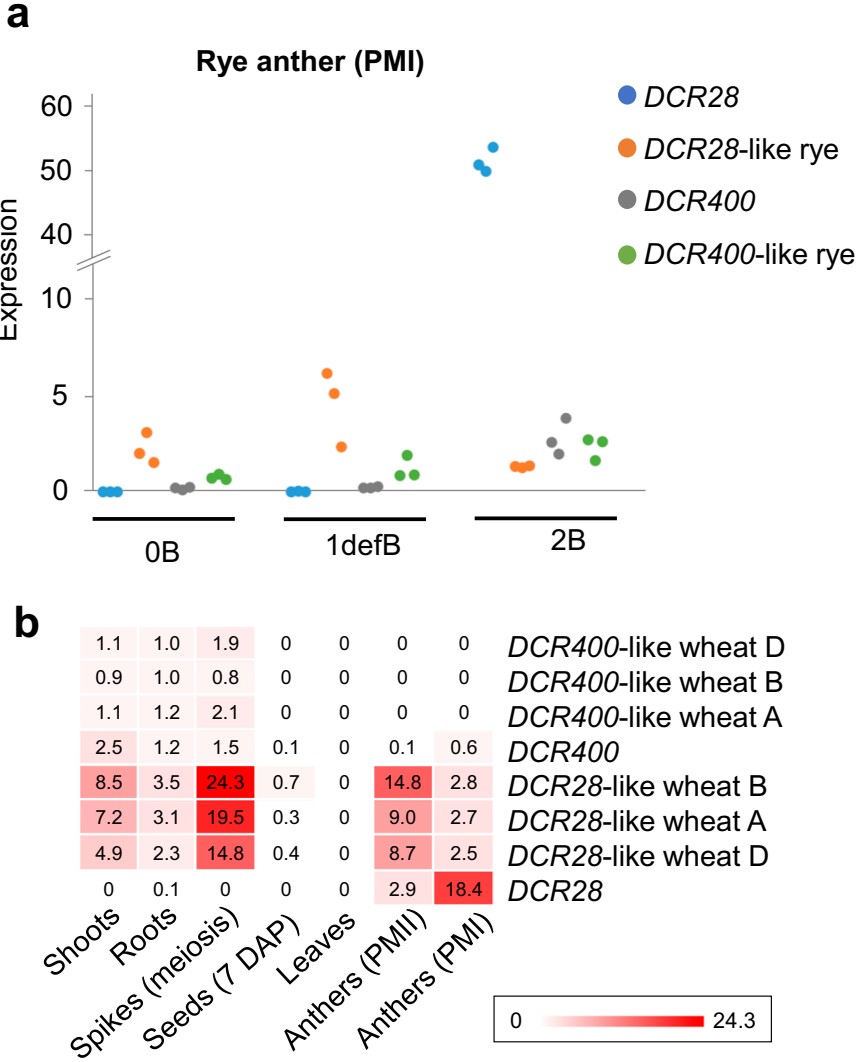

**Fig. 6 | The transcriptional activity of *DCR28* and its paralogous genes differs, while the transcriptional activity of *DCR400* and its paralogous genes are similar. a** The expression of *DCR28* (A) and its rye A-paralog (*DCR28*-like rye, SECCE7Rv1G0479160) (B), *DCR400* (C) and its rye A-paralog (*DCR400*-like rye, SECCE1Rv1G0060580) (D) in rye with 0B, 1 deficient B (defB), and 2B chromosomes during the first pollen mitosis (PMI). Expression calculation method: transcripts per million (TPM). **b** The average expression of *DCR28* and the expression of *DCR400*, *DCR28*-like, and *DCR400*-like genes in wheat in different tissues of wheat + 2Bˢ:

Shoot apical meristems, root apical meristems, spike-young spikes undergoing meiosis, seeds (7 days after pollination), leaf-young and old leaves, and anthers undergoing second pollen mitosis (PMII) and the first pollen mitosis (PMI). Expression calculation method: transcripts per million (TPM). *DCR400*-like wheat D: TraesCS1D03G0983600, *DCR400*-like wheat B: TraesCS1B03G1206100, *DCR400*-like wheat A: TraesCS1A03G1024600. *DCR28*-like wheat D: TraesCS4D03G0484200, *DCR28*-like wheat B: TraesCS4B03G0550000, *DCR28*-like wheat A: TraesCS4A03G0224100. Source data are provided as a Source Data file.

tissues including PMI and PMII anthers (Fig. 6b). Thus, expression dynamics definitively differ between *DCR28* and the A chromosome-encoded *DCR28*-like genes of rye and wheat. In contrast, no significant DGE is evident between *DCR400* and *DCR400-like wheat A, B, D* (Fig. 6b).

## The phylogeny of *DCR28*

*DCR28* is classified as a member of the microtubule-associated Futsch-like protein family (PTHR34468) by PANTHER. To better understand its evolution, a phylogenetic tree of *DCR28*-like genes was built based on 434 protein sequences of 325 species obtained from the NCBI database. Members of this gene family were detected throughout the major plant groups from algae to angiosperms. Rooting the tree with *Chara brownii*, an alga representing the phylogenetically earliest branch in our dataset, we found that an early gene duplication resulted in two major groups within the *DCR28*-related genes (Fig. 7). This duplication must have happened at, or after the origin of the

angiosperms, as in algae, sporophytes, and gymnosperms we found only one type. However, as gymnosperms possess a copy that groups at the base of type 1 of the gene family (blue in Fig. 7) and, at least in some analyses, the fern sequence of *Adiantum nelumboides* groups at the base of type-2 sequences (Supplementary Fig. 17) the duplication could have also happened within early seed plants. Not all angiosperm species in our dataset carry both types of *DCR28*-related proteins. It seems that type-2 sequences (red in Fig. 7) were particularly lost in many branches of angiosperms. The *DCR28* sequences of rye and *Ae. speltoides* group with *Triticum*- and Triticeae-derived genes within the grass clade of type 2, placing its origin within this group.

## DCR28 is a microtubule-associated protein

To determine whether DCR28 and its rye A chromosome paralog DCR28-like are related to chromosome segregation, we fused the Enhanced Yellow Fluorescent Protein (EYFP) to their N-termini and analyzed their subcellular localization in transient expression assays

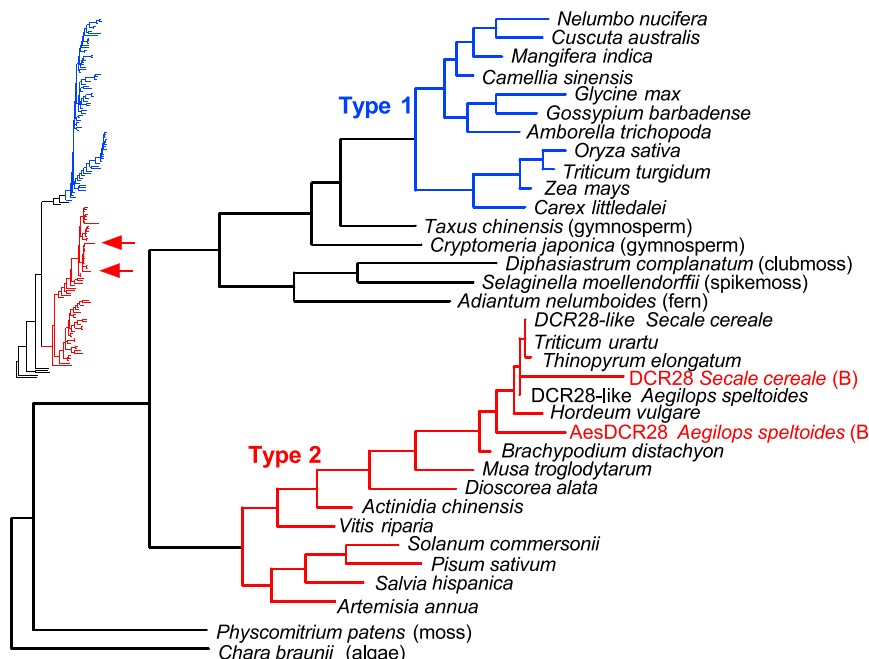

**Fig. 7 | Gene tree indicating phylogenetic relationships of the Futsch-like proteins.** Maximum-likelihood phylogenetic tree of representative protein sequences rooted with *Chara braunii*. In blue and red the two branches of the protein family are indicated, which resulted from a gene duplication in early angiosperms or seed plants. (B) indicates the DCR28 from B chromosomes. The insert to the left provides the tree topology of an extended analysis, including 139 sequences (Supplementary Fig. 17), with an arrow indicating the position of the two DCR28 sequence.

under the control of the CaMV35S promoter in *Nicotiana benthamiana* (*N. benthamiana*) leaves. Both showed fiber-like signals on the cytoskeleton compared to the EYFP control (Fig. 8a). Independent co-infiltration with mCherry-Microtubule Binding Domain (mCherry-MBD)[46] and mCherry-Actin Binding Domain 2 (mCherry-ABD2)[47] revealed that DCR28 and DCR28-like rye are associated with microtubules rather than with actin (Supplementary Fig. 18). However, among cells that showed microtubule-like signals, DCR28 alone appeared to encourage the formation of bundled fibers, which were observed in 28% (94/336) of cells expressing EYFP-DCR28, but in no cells expressing EYFP-DCR28-like rye (Fig. 8a).

To ascertain whether DCR28 and DCR28-like rye relate to cell division, we employed the cell division-enabled leaf system[48] in a stably transformed histone CFP-H2B reporter line of *N. benthamiana*. DCR28 and DCR28-like rye localize to the spindle during cell division, and their signals are identical to the microtubule during cell division (Fig. 8b and Supplementary Fig. 19). To find differences between DCR28 and DCR28-like rye during the cell division, we fused the fluorescent protein mCherry to the N-termini of DCR28 and DCR28-like rye, and co-infiltrated EYFP-DCR28-like rye and EYFP-DCR28, respectively. No difference was observed between DCR28 and DCR28-like rye during prophase to telophase (Supplementary Fig. 20). We conclude that DCR28 and DCR28-like rye are microtubule-associated proteins and only DCR28 promotes microtubule-bundling. Both participate in cell division and are likely involved in chromosome segregation.

## Discussion

We report the identification of rye B chromosome drive-controlling candidate genes. Although the post-meiotic irregular segregation of the rye B was reported early last century[19], and the B repeat-enriched region of the long chromosome arm was identified as a trans-active region controlling drive[20–24], the identification of genes controlling this process was lagging behind. The non-Mendelian segregation of the rye B chromosome prevented the application of a recombination-based method for gene mapping and identification. To circumvent this, we

utilized a combination of complementary genomic, transcriptomics, and chromosomal techniques. In addition, the ability of the rye B chromosome to drive in the genomic background of wheat as in rye[21,29], and the availability of wheat lines possessing B variants with and without the driving ability[21] were elemental to the success of the project. Further, the identification of rye B sequences and transcripts was made easier by the lower sequence similarity between the wheat A chromosomes and the rye B than rye A and B chromosomes.

Out of the 1292 transcriptionally active genic B sequences found in the wheat background, a multi-step comparative transcriptome analysis together with physical mapping, was utilized to identify potential genes governing the drive of the B chromosome. The initially identified 533 upregulated B-encoded genes were finally reduced to five candidates encoded by the B DCR. Among them are two candidates of interest: *DCR28*, which we proved to be a microtubule-associated protein related to cell division, and *DCR400*, classified as DNA helicase likely related to DNA duplex unwinding and sister chromatid cohesion. Likely because of the independent origin and different mechanisms of the rye B and maize B chromosome drive, none of the 34 candidates for a trans-acting, protein encoding drive gene of the maize B chromosome[10] resembles any of the identified rye B drive candidate genes.

Although we applied the top techniques used for genome assemblies (PacBio HiFi, ONT, Hi-C, and optical mapping), high sequencing depth (~44× PacBio reads for the rye B chromosome), new genome assembly algorithms like those used for the human telomere-to-telomere project[49,50], and contiguous chromosome-scale assemblies of all wheat A chromosomes were achieved, the assembly of a ~430 Mb large B chromosomal pseudomolecule was only possible after employing, in addition, FISH-mapped B sequences as barcodes for scaffolding. We found that some B satellite repeat arrays (E3900, CL11, Sc26c38) were underrepresented in the rye B pseudomolecule, and therefore, the majority of sequence gaps might be attributed to assembly breakdown in long stretches of satellite repeats. In addition, the difference between assembly size and flow cytometric genome size

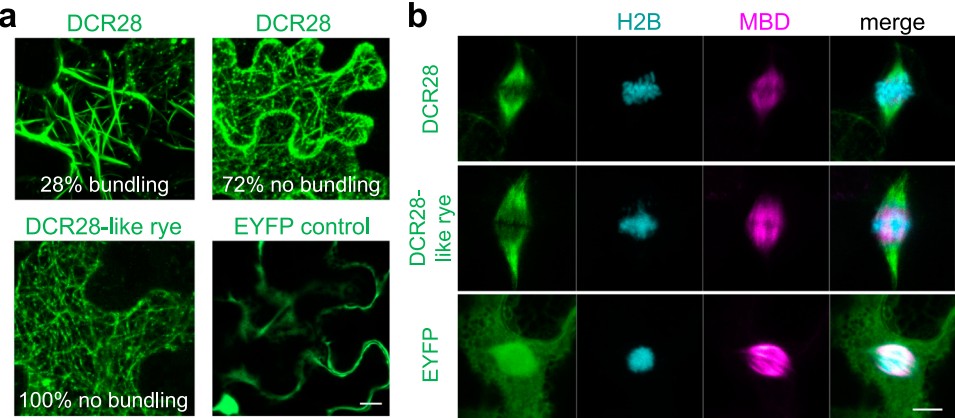

**Fig. 8 | DCR28 is a microtubule-associated protein.** Transient co-overexpression of EYFP-DCR28 and EYFP-DCR28-like rye (in green) and mCherry-MBD (in magenta) in *N. benthamiana* stably expressing histone H2B-CFP (in blue). **a** Overexpression of EYFP-DCR28 results in 28% tubulin fiber bundling at interphase (*n* = 336). Overexpression of DCR28-like rye results in no bundling of fibers (*n* = 411). **b** Note spindle localization of EYFP-DCR28 and EYFP-DCR28-like rye during metaphase. The experiment was independently repeated twice with similar results. Bars = 10 μm.

estimates might also account for missing sequences[51]. A similar assembly challenge was reported for the pseudomolecules of the germline-restricted chromosomes (GRCs) of songbirds, which were highly fragmented and only accounted for 36–75% of the estimated sizes[52]. On the other hand, the sequence assembly of the maize B pseudomolecule covered 76% of its size (106.6/141 Mb)[10].

We consider the *DCR28* gene family as a candidate to control in trans the drive of the rye B chromosome because of its B-specific and high first pollen mitosis-specific transcription activity, the post-meiotic division, where the drive of the B occurs irrespectively of the host species. In addition, *DCR28* is localized in the trans-active DCR of the rye B chromosome, is conserved in the B chromosomes of cultivated and weedy rye and *Ae. speltoides*, and expression of an DCR28 reporter shows colocalization with mitotic tubulin fibers and triggered tubulin bundling. A further interesting observation is that the *DCR28* family consists of 15 members derived from an evolutionarily conserved single-copy A chromosome paralog. Also in other species, genes encoding chromosome drive occur in multiple copies like, *Kindr* (8 copies) *in Zea mays*[53], *R2d2* (>30 copies) in mice[54], *Dxl* (5-12 copies) in *Drosophila*[42], and *wtf* (up to 42 copies) in *Schizosaccharomyces*[55]. De Carvalho et al. proposed that when duplications of drive-controlling genes occur, they must be selected for increased strength of the drive mechanism. Then, once duplicated, the drive-controlling genes can be reborn[55]. Indeed, a correlation between the copy number of driver genes and the frequency of drive has been demonstrated for *R2d2*[56]. It was proposed that additional *Dxl* copies may have increased the strength of the drive or facilitated the escape from endogenous small interfering RNAs (esiRNA)-mediated silencing by autosomal suppressors[42].

Although, according to gene annotation, *DCR400* is likely involved in the extended cohesion of the sister chromatids, the high similarity (98%) between *DCR400* and its paralog on the A chromosome suggests that *DCR400* is a young gene which the B chromosome 'gained' recently, in contrast to its long independent evolution (more than one million years). Moreover, it is also expressed in tissues where no drive of the B chromosome occurs. On the other hand, the B chromosomal drive process is likely not regulated by a single gene. Additional genes, such as *DCR400* in conjunction with *DCR28*, may aid in the regulation of the B drive. It is conceivable that the gene composition needed for drive evolved successively, with *DCR400* emerging as a component that adds to drive of rye in more recent evolution.

In order to drive a specific chromosome, besides the trans-active element, a cis-chromosome-specific component might exist. In maize, for example, the drive of abnormal chromosome 10 (Ab10) depends on the interaction between its heterochromatic knobs and the kinesin-14 motor *Kindr*[53]. We speculate that the rye B pericentromere-specific repeat CL11[20] or another B repeat, directly or indirectly in combination with a repeat-specific binding protein, acts as cis-element of B chromosome nondisjunction and drive. A specific composition of the B (peri)centromere has also been reported in *Ae. speltoides*[39] and maize[57]. In all three species, B-specific nondisjunction results in drive. In rye, the cohesion between B sister chromatids during first pollen mitosis is likely stronger than the microtubule traction force required for the separation of chromatids. The suggested chromosomal position of the B drive cis-element is supported by the analysis of the non-driving rye B/wheat A translocation chromosome B$^{s-10}$, which contains almost half of the long B arm, including the DCR[21].

Based on our results, we propose that the high activity of the B-encoded DCR28 family during the first pollen mitosis modulates the microtubule structure and bundling of tubulin fibers occurs as identified in *N. benthamiana*. Consequently, the forces required to separate B sister chromatids are insufficient, resulting in nondisjunction and lagging of B sister chromatids resulting in B chromosome drive. In contrast, the sister chromatids of A chromosomes disjoin because the forces required to separate A sister chromatids are above the required threshold. Alternatively, DCR28-enriched microtubules interact with B sister chromatids and antagonize the pulling force of kinetochore-microtubules from both spindle poles and finally cause the nondisjunction of the B chromosome. Or, the microtubule binding of DCR28 during the first pollen mitosis leads to an erroneous microtubule attachment on the B centromere from different orientations, resulting in a merotelic attachment and lagging B chromosomes. An almost comparable situation has been reported for R2d2, a 127 kb monomer repeat located on mouse chromosome 2[58]. The R2d2-containing chromosome shows lagging during anaphase of female meiosis to preferentially remain in the egg. It is suspected that R2d2 must interact with some structures, like the spindle, to slow its poleward movement during anaphase.

In concurrence with the first rye B chromosome sequence characterization that was based on short sequence reads derived from flow-sorted B chromosome[13], our analysis demonstrates that the rye B represents a complex mosaic of rye A chromosome- and organellar-derived sequences. A comparable sequence composition was reported for the Bs of *Ae. speltoides*[59] and maize[10], for the B-like GRCs of songbirds[52,60] and the paternal sex ratio chromosome, a B chromosome in the jewel wasp *Nasonia*[11].

We suspect that the A-located single-copy gene *DCR28*-like rye underwent neo-functionalization and amplification during the early

stage of B chromosome formation, and the resulting *DCR28* family became essential for the drive process of the B. After the sub-chromosomal regions around the *DCR28* locus lost synteny because other A-derived genes intermingled with *DCR28* members derived from other A chromosomes like 4R and 5R.

## Methods

### Plant materials

Cultivated rye (*Secale cereale* L. subsp. *cereale*) of the Japanese JNK strain without and with additional standard or deficient rye Bs (defB), which lost the ability to drive due to the loss of the nondisjunction control region[61] and weedy rye containing B chromosomes collected from Pakistan (*Secale cereale* L. subsp. *segetale*, no. 34)[28] and Afghanistan (*Secale cereale* L. subsp. *afghanicum*[27], were grown under long-day conditions of 16 h light at 18 °C and 8 h dark at 15 °C, 50% relative humidity, and 100–120 µmol m$^{-2}$ s$^{-1}$ light intensity. They were vernalized at 4 °C for 2 months at the third leaf stage and then moved back to the greenhouse.

Wheat (*Triticum aestivum* L., cv Chinese Spring) without and with different variants of rye B chromosome (B$^s$, B$^s$−8, B$^k$, B$^k$−1, B$^k$−2 [21], and B$^k$−3) were grown under greenhouse conditions with a 16 h photoperiod (21 °C day/17 °C night, 50% relative humidity, 100–120 µmol m$^{-2}$ s$^{-1}$, light intensity). The B$^s$ chromosome of the wheat-rye B addition line was initially introduced into a Nepalese strain of wheat from a spring rye variety from Transbaikal, Siberia[29] and then transferred into the wheat genotype 'Chinese Spring'[21]. The B$^k$ chromosome was derived from cultivated rye in Korea, and it was introduced into wheat cv. Chinese Spring by Niwa et al.[62]. The B chromosome variants B$^s$−8, B$^k$−1, and B$^k$−2 were created by the application of a gametocidal system, which resulted in the generation of deficient B chromosomes like, terminal deletions and translocations with wheat chromosomes[21]. The B$^k$−3 was identified when we screened the selfing progenies of wheat with B$^k$−2 by FISH.

*Ae. speltoides* Tausch with or without B chromosomes from Tartus, Syria (K2 and D2), Katzir, Israel (H7), Ramat Hanadiv, Israel (P12) were grown under long-day conditions of 16 h light at 25 °C and 8 h dark at 16 °C, 40% relative humidity, and 50–60 kilolux light intensity. They were vernalized at 4 °C for 4 weeks at the third leaf stage and then moved back to the greenhouse.

Wild type and transgenic *N. benthamiana* Domin expressing the reporter construct 35S::CFP-histone H2B[63] were grown under greenhouse conditions with a 16 h photoperiod (22 °C day/18 °C night, 50% relative humidity, 100–120 µmol m$^{-2}$ s$^{-1}$ light intensity).

### Preparation of chromosome spreads

To prepare mitotic chromosomes, seeds were germinated on wet filter paper at room temperature (RT) for 2–3 days. Excised roots were treated with ice-cold water for 24 h for cell-cycle synchronization, fixed in ice-cold 90% acetic acid for 10 min on ice or 3:1 ethanol:acetic acid for 1–3 days at RT, and stored in 70% ethanol at −20 °C. Roots were treated with 45% acetic acid between 10 and 120 min before being transferred onto a glass slide. Meristematic cells were then isolated in a droplet of 45% acetic acid under a coverslip. After 2 s of treatment over an alcohol burner, the meristem was squashed between the slide and coverslip. Finally, the coverslip was removed after freezing it in liquid nitrogen, and the slide was kept in 99.8% ethanol at −20 °C until use.

To prepare pachytene chromosomes, spikes of wheat with rye B$^s$ were collected and fixed in 3:1 ethanol: acetic acid for 3 days following emergence of 2/3 of the flag leaf. Three anthers per sample were then transferred to a 0.2 mL tube containing 10 µL 45% acetic acid and mixed to create a homogeneous cell suspension. Subsequent slide mounting and storage followed the steps for mitotic chromosomes.

### Standard fluorescence in situ hybridization, probe generation, and microscopy

The PCR products and plasmids used as FISH probes were fluorescence labeled by nick translation (NT Labeling Kit, Jena Bioscience). The repeats Sc26c38 and Sc9c130 were obtained by PCR using the primers described by Klemme et al.[26]. The gene fragment of *DCR28* was obtained by PCR using the primer pair DCR28-3 (Supplementary Table 4). pTZE3900, containing a 3.9 kb long fragment of the repeat E3900[64], and pUC119-Revolver, containing an 89 bp long sequence of the rye genome-specific repeat Revolver[65] were used. In addition, four oligonucleotides (30–40 nt, Oligo-D1100-mix) with fluorescein isothiocyanate (FITC) at their 5′-end were designed to target the sequence of the repeat D1100[66]. The oligo probes 5S rDNA[67] were modified with 5-TAMRA (5-carboxytetramethylrhodamine) at their 5′-end. The sequences of primers and oligo probes are listed in Supplementary Table 4. Chromosomes were denatured in a NaOH–70% ethanol solution (6 mg/mL) for 5 min at RT, then washed, dehydrated in a series of increasing ethanol concentrations (70%, 90%, and 99.8%) for 5 min each, and air-dried. For each FISH probe, 1 µL of probes (10 mM for oligo probes; 50–75 ng/µL for probes generated by nick translation) was added to 10 µL hybridization mixture, denatured at 99 °C for 10 min and stored at immediately −20 °C until use. The mixture was added to the air-dried slides, sealed with coverslips, and incubated in a moist chamber at 37 °C for 12–24 h[68]. After hybridization, slides were washed in 2X SSC for 20 min at 58 °C and distilled water at RT for 2 min, and air-dried. Finally, 8 µL 4′,6-diamidino-2-phenylindole (DAPI) solution (1 µg/mL, DAPI/antifade solution) in antifade was added to each slide and sealed with a coverslip. Microscope images were taken using an Olympus BX61 fluorescence microscope equipped with an ORCA-ER CCD camera (Hamamatsu) and a deconvolution system. Images were analyzed using the cellSens Dimension software (Olympus, v1.11) and Adobe Photoshop (v13.0).

### Pollen-FISH

Mature pollen was fixed in ice-cold 90% acetic acid for 20 min and maintained in 70% ethanol at −20 °C. The protocol for suspension pollen-FISH[69,70] was adapted as following. After centrifugation (12,000 × *g* for 1 min), the ethanol was removed, and the pollen pellet was rinsed with 1 mL 10 mM HCL three times by vortexing and centrifugation (12,000 × *g* for 1 min) at RT. Pepsin solution (80 µL at 20 mg/mL, dissolved in 10 mM HCL) was added, and the solution was incubated at 37 °C for 30 min. The pollen was rinsed with 2X SSC by vortexing and centrifugation (9500 × *g* for 1 min), and the pollen pellet was denatured in 100 µL NaOH−ethanol (6 mg/mL) for 5 min at RT, pollen was rinsed with 2X SSC twice by vortexing and centrifugation (9500 × *g* for 1 min). Oligos were combined and diluted in hybridization mix (10 mM oligo-D1100-mix; 10 mM oligo-5S rDNA-mix), denatured at 99 °C for 10 min, and maintained at −20 °C for 10 min[68]. The oligo mixture was added to the pollen pellet and incubated in the dark for 20–24 h at 37 °C. Following hybridization, the pollen was washed twice in 2X SSC at RT by vortexing and centrifugation (12,000 × *g* for 1 min), and incubated for 30 min at 45 °C in 2X SSC. The supernatant was discarded, the pollen pellet resuspended in 15 µL DAPI solution (1 µg/mL DAPI/antifade solution) dropped onto slides, and sealed with a coverslip. Slides were incubated in darkness at RT overnight and maintained at 4 °C until microscopy. To achieve an optical resolution of ~120 nm applying a 488 nm excitation, we performed spatial structured illumination microscopy (3D-SIM) using a 63×/1.40 objective of an Elyra PS.1 super-resolution microscope system and the software ZENBlack (Carl Zeiss GmbH). Image stacks were captured separately for each fluorochrome using 561, 488, and 405 nm laser lines for excitation and appropriate emission filters[71,72]. A 3D-image stack was used to generate Supplementary Movie 1 using the Imaris 9.7

(Bitplane) software. Frequency of B chromosome drive was calculated using the following equation:

Frequency of B chromosome drive = number of pollen showing B-signals only in sperm nuclei/number of pollen showing B-signals in any type of nuclei (1)

## DNA and RNA isolation, cDNA synthesis, and short-read sequencing

Leaf segments (1–3 cm) from rye and wheat samples were frozen in tubes by immersion in liquid nitrogen pulverized by bead beating in a vibrating mill (MM400, Retsch, frequency: 30/s, 1 min). Extraction buffer (1.21 g Tris, 4.09 g NaCl, 1.86 g EDTA, diluted in 100 mL H$_2$0; 1.2 mL/sample) was added to each tube and incubated for 15 min at 65 °C. Tubes were allowed to cool for 1 min at RT. Chloroform: isoamyl alcohol (24:1, 600 μL) was added, and the solution was manually agitated. The homogenate debris was pelleted by centrifugation (13,000 × $g$ for 2 min), and the supernatant containing DNA was moved to a fresh tube. The DNA was precipitated in 700 μL isopropanol, pelleted, and washed twice in 70% ethanol by repeated vortexing and centrifugation (13,000 × $g$ for 2 min). The pellet was air-dried until all ethanol was evaporated, and the DNA was dissolved in 50 μL of dH20. The DNA concentration was determined by spectrophotometry (Thermo Scientific NanoDrop One) and the Qubit device (DNA HS assay kit, Thermo Fisher Scientific Inc, Waltham, MA, USA).

RNA from six tissues (root meristem, PMII anther, leaf, spike undergoing meiosis, immature shoot, and immature seeds 7 days after pollination) of wheat with 2B$^s$ were collected (Supplementary Data 5). For comparative transcriptome analysis, RNA-seq data of PMI wheat + 0B/2B/4B and rye + 0B/2B[16], and of PMI anthers of wheat + 1B$^s$–8/1B$^k$–1/1B$^{k-2}$ and rye + 1defB were collected (Supplementary Data 5). Anthers undergoing different stages of development (meiosis, PMI, and PMII) were classified by visual inspection under a light microscope. Anthers (undergoing PMI or PMII) or whole young spikes (undergoing meiosis) were immediately frozen in liquid nitrogen and stored at −80 °C. Total RNA was extracted using the Spectrum™ Plant Total RNA-Kit (Sigma-Aldrich). RNA samples were treated with DNA-free DNase before cDNA synthesis following the on-column DNA removal protocol (RNase-Free DNase I Kit, Norgen Biotek Corp.).

RNA was quantified using the Qubit device (RNA HS assay kit, Thermo Fisher Scientific Inc, Waltham, MA, USA). Total RNA quality was verified by determining the RNA Integrity Number (RIN) using the Agilent Technologies 2100 Bioanalyzer (Agilent, Santa Clara, CA, USA). Sequencing libraries were prepared using the Illumina TruSeq RNA Sample Preparation Kit v2 (Illumina, Inc., San Diego, CA, USA) followed by size selection of the pooled samples by agarose gel electrophoresis (size range: 320–420 bp. Libraries were quantified by qPCR[73] and sequenced on the Illumina HiSeq 2500 device (paired-end, 2 ×101 cycles, rapid run mode, onboard clustering; DNA Sequencing Service of the IPK Gatersleben, Germany) according to the manufacturer's directions (Illumina, Inc., San Diego, CA, USA). The raw RNA-seq reads were trimmed using Trimmomatic (v0.38.1[74], SLIDINGWINDOW:4:20, MINLEN:80). For short-read sequencing, the genomic DNA of wheat + 3B$^k$–2, wheat + 2B$^k$–3, and wheat +2 B$^s$ were sequenced using the DNBSEQ sequencing platform at BGI Genomics (Hong Kong, China). About 36 Gb paired-end 150 (PE150) data were generated for each sample.

## PacBio sequencing

High-molecular-weight (HMW) DNA of wheat cv. Chinese Spring possessing ~6 rye standard B$^s$ (Supplementary Fig. 4) was isolated from leaves using the NucleoBond HMW DNA kit (Macherey Nagel, Germany), and quality was assessed using the Fragment Analyzer device (Agilent Technologies Inc, CA, USA). DNA was quantified using the Qubit dsDNA High Sensitivity assay kit (Thermo Fisher Scientific, MA, USA). HiFi libraries were prepared from 15 μg HMW DNA according to

the 'Procedure & Checklist – Preparing HiFi SMRTbell® Libraries using SMRTbell Express Template Prep Kit 2.0' manual (PN 101-853-100 Version 03, Pacific Biosciences of California Inc., USA) with an initial DNA fragmentation (speed 32) performed using the Megaruptor 3 device (Diagenode, Belgium) and final library size fractionation by SageELF (Sage Science, USA). The size of the final libraries was measured using the Fragment Analyzer device (Agilent Technologies Inc, CA, USA). Polymerase-bound SMRTbell complexes were formed according to standard protocols (Pacific Biosciences of California Inc., USA). Sequencing (HiFi CCS) was performed using the Pacific Biosciences Sequel IIe device (30 h movie time, loading concentration 45–70 pM, 4 h pre-extension time, diffusion loading, mean insert length according to SMRT link raw data report between 16.4 and 19 kb) following standard manufacturer's protocols (Pacific Biosciences of California Inc., Menlo Park, CA, USA) at IPK Gatersleben and Pacific Biosciences Revio platform at BGI Genomics (Hong Kong, China). The Sequel IIe generated ~92 Gb data and the Revio generated ~166 Gb data.

## Nanopore sequencing

Nanopore sequencing was performed with HMW DNA extracted from purified leaf nuclei of the same wheat cv. Chinese Spring cultivar as used as for PacBio sequencing[75]. The quality of the DNA preparations was checked by field inversion gel electrophoresis to ensure that the DNA fragment size was >100 kb. Ultra-long read sequencing libraries were prepared from 35–40 μg of HMW DNA using the SQK-ULK001 kit (Oxford Nanopore Technologies) according to the manufacturer's instructions. The libraries were sequenced on two FLO-PRO002 flow cells on the PromethION sequencer and yielded 44.3 and 42.0 Gb of sequence data with read N50 value of 64 and 44 kb, respectively. In total 69.0 Gb pass Quality Score 10 (Q10).

## Chromosome conformation capture sequencing

Hi-C sequencing libraries were generated from leaves of wheat with two rye B chromosomes[76] using *Dpn*II enzyme, and were sequenced using the NovaSeq6000 device (Illumina Inc., USA) at IPK Gatersleben. Approximately 173 Gb were generated, and ~148 Gb true Hi-C reads were used for scaffolding after filtering.

## Optical genome mapping

B chromosomes were purified by flow cytometry from the same line as for PacBio and Nanopore sequencing (cv. Chinese Spring with six rye B$^s$ chromosomes). Five million mitotic chromosomes were flow-sorted after cell-cycle synchronization[77]. The purity of the sorted fraction was estimated as 70% by FISH, and the major contaminants were wheat chromosomes 1D and 6D. Sorted chromosomes were embedded in agarose plugs and treated with proteinase K[78], resulting in 640 ng HMW DNA. After release from the agarose plugs, the DNA was directly labeled at DLE-1 recognition sites (CTTAAG motif) following the standard Bionano Prep Direct Label and Stain (DLS) Protocol (Bionano Genomics, San Diego, USA), and analyzed using the Bionano Genomics Saphyr platform. Data were collected for all single molecules >150 kb, totaling 816.7 Gb and further filtered based on molecule maximum fluorescence intensity and length. The resulting dataset, consisting of 92 Gb of molecules >300 kb (N50 = 428.3 kb) with a maximum fluorescence intensity of 2000, comprised 115 equivalents of the rye B chromosome. This dataset was used to de novo assemble an optical genome map (OGM) (Supplementary Table 5) using the Bionano Solve software (version 3.6.1_11162020; parameters 'optArguments_nonhaplotype_noES_noCut_DLE1_saphyr.xml'). The resulting OGM consists of 262 contigs with a total length of 411.4 Mb. To discriminate residual wheat map contigs, the map assembly was aligned to the reference genome of wheat cv. Chinese Spring (IWGSC RefSeq2.1). A total of 172 short OGM contigs, representing 22.8% of the entire OGM assembly length (93.6 Mb), aligned to wheat chromosomes 1D and 6D. The

remaining 90 contigs, having a cumulative length of 317.8 Mb and N50 of 8.5 Mb, belonged to the rye B chromosome.

## Whole-genome assembly and scaffolding

PacBio HiFi reads and Nanopore ultra-long reads of wheat cv. Chinese Spring possessing ~6 rye standard Bˢ were assembled into contigs with hifiasm[49,79] (v0.19.3-r572; parameters: -l 0 -D 20, Ultra-long ONT integration). The coverage of HiFi reads on each contig was extracted from the Graphical Fragment Assembly (GFA) file output by hifiasm. Contig statistics were calculated with Quast[80] (v2.3) and gene content completeness was evaluated with Benchmarking Universal Single-Copy Orthologs (BUSCO) (v4.1.2; dataset: Viridiplantae Odb10)[81]. The Arima Genomics mapping pipeline (https://github.com/ArimaGenomics/mapping_pipeline) was used to process the Hi-C data, including read mapping to the contigs, read filtering, read pairing, and PCR duplicate removal, and scaffolding was performed using YaHS (v1.2a.2)[82]. Hi-C contact maps were generated using Juicebox (https://github.com/aidenlab/Juicebox).

To evaluate the quality of the assembly and scaffolding, synteny between the 21 largest scaffolds (expected to correspond to the 21 bread wheat chromosomes) and the IWGSC RefSeq v2.1 assembly of wheat cv. Chinese Spring was established by sequence homology. Hundred bp sequences spaced 10-kb apart on each scaffold were aligned to the IWGSC RefSeq v2.1 assembly via blastn (blast program: megablast, v2.10.1). The results were visualized by Dot plot using R (v4.01).

## Scaffolding of the rye B chromosome

The OGM was used together with rye B-assigned contigs as input files for the automated hybrid scaffolding (HS) pipeline integrated in Bionano Solve (version 3.6.1.11162020). The HS pipeline was run with default configuration and a 'Resolve conflict' option for conflict resolution. Flagged conflicts between sequences and the optical map were manually reviewed, and the HS pipeline was re-run, resulting in a HS assembly consisting of 18 hybrid scaffolds with a cumulative length of 377.1 Mb (Table S1). No OGM contig derived from wheat chromosomes 1D and 6D was integrated into the hybrid scaffolds.

Scaffolding based on Hi-C and the OGM was complemented by a cytogenetics approach. Bilby (GenBank: AF245032.1), CL11 (GenBank: JQ963576.1), Sc55c1 (GenBank: KC243248.1), Sc63c34 (GenBank: KC243249.1), D1100 (GenBank: KC560866.1), Sc26c38 (GenBank: KC243242.1), E3900 (GenBank: AF222021.1), Sc9c130 (GenBank: KC243235.1), and mitochondrial DNA (GenBank: AP008982.1) and chloroplast DNA (GenBank: NC_021761.1) were aligned to the contigs using BLASTn (blast program: megablast, v2.10.1). The abundance of each repeat on the contigs was quantified in 1-Mb windows (Supplementary Data 3). Links from Hi-C data and optical mapping data were used to combine the contigs into super-scaffolds (Supplementary Data 2, 3). The super-scaffolds were ordered into a pseudomolecule based on their repeat content and FISH results from Klemme et al.[26]. A fasta file was generated with agptools (https://warrenlab.github.io/agptools/). The distribution of the repeats on the B pseudomolecule (chrB) was visualized by pyGenomeTracks (v3.8)[83]. To determine the DCR, DNBSEQ short-read data of wheat + 3Bᵏ–2, wheat + 2Bᵏ–3, and wheat + 2Bˢ were aligned to the reference genome of the 21 wheat large scaffolds, B pseudomolecule and unassigned rye B-like contigs using bowtie2 (v2.5.0, default)[84]. The alignments on the B pseudomolecule and unassigned rye B-like contigs were extracted via samtools (v1.9)[85] and visualized by pyGenomeTracks (v3.8). LAI was used to assess the assembly quality of the B pseudomolecule[33].

## Comparing the assembled size of repeats to unassembled short-read data

To estimate the actual proportion of assembled B-located repeats, 17.2 Gb (representing 1x genome coverage) short-read DNBSEQ

sequencing data of wheat with 2Bˢ (single-end, 150-bp) was selected. The B pseudomolecule was cut into 150 bp sequences. Both data were aligned to the wheat genome (IWGSC RefSeq v2.1 assembly) using bowtie2 (v2.5.0, default) and unaligned reads were written to separate files. Organellar DNA was not able to be compared since they also exist in cytoplasm. Sc55c1 was not analyzed as it has similar sequences in the standard wheat genome. Therefore, the unaligned DNBSEQ short reads and unaligned 150-bp pseudomolecule sequences were aligned to the remaining seven repeats Bilby, CL11, Sc63c34, D1100, Sc26c38, E3900, and Sc9c130 using bowtie2 (v2.5.0, default). The numbers of aligned reads of each repeat were used for calculation. The proportion of the assembled size of each repat is calculated as following:

$$\text{The proportion of the assembled size of each repeat} = \text{number of 150-bp pseudomolecule sequences/number of DNBSEQ short reads}\quad(2)$$

## Transcriptome assembly and differential expression analysis

Cleaned RNA sequence reads (Supplementary Data 5) were mapped to a hybrid reference, including the rye B contigs and the genome of wheat cv. Chinese Spring (IWGSC RefSeq v2.1), with HISAT2 (v2.2.1, default parameters)[86]. The alignment was processed to produce a gene feature annotation with StringTie (v2.1.1, default parameters)[87]. A set of non-redundant transcripts was generated using gffread (v0.12.6)[88]. TransDecoder (v5.5.0) was used to annotate coding regions within transcripts.

Differential expression analysis of plants with a wheat genetic background was performed against a reference transcriptome featuring the rye B transcripts combined with the annotated transcriptome of wheat cv. Chinese Spring (IWGSC RefSeq 2.1, using both HC and LC genes). The corresponding analysis of plants with a genetic rye background substituted the rye 'Lo7' genome for the wheat genome[35]. Salmon (v3.0, default parameters)[89] was used to estimate the abundance of each transcript from each tissue:sample combination based on the RNA-seq reads. DESeq2 (v1.34.0)[90] was used to compare the expression of each transcript during PMI between genotypes with and without the DCR of the rye B (adjusted $p$ value < 0.01, fold change > 2) (Fig. 4a). Transcripts showing differential expression in all 11 comparisons were selected as candidate trans-acting factor(s) influencing PMI nondisjunction of the rye B.

## Analysis of the differentially expressed candidates and PCR-based mapping

DEGs were clustered using CD-HIT-EST (similarity threshold: 0.8, v1.3)[91]. The PANTHER classification system (http://www.pantherdb.org) was used to infer the likely functional roles of the translated proteins. To find paralogous A chromosome genes, the sequences of the candidates were aligned to the transcriptome of the rye cv. Lo7[35] (blast program: megablast, v2.10.1). Where PANTHER can not classify the translated proteins, the homology-based functional annotations from rye cv. Lo7[35] were assigned their homologous B chromosome-located candidates. B chromosome-specific primers were generated using the WheatOmics PrimerServer tool[92,93] compared to wheat IWGSC RefSeq v2.1 assembly and rye 'Lo7' reference genome[34,35] (Supplementary Table 4). Genomic DNA of plants with or without DCR of the rye B chromosome were used as templates. GoTaq DNA Polymerase (Promega) was used for PCR following the protocol recommended by Promega, and the PCR products were checked using 1% agarose gels.

## De novo assembly of transcriptome of Ae. speltoides +B

Approximately 9.5 Gb RNA-seq data were produced from anthers (undergoing the first pollen mitosis) of Ae. speltoides with 3 B chromosomes (Genotype K2 from Tartus, Syria). Trinity (v2.4.0, default parameters)[94] was used to de novo assemble the transcriptome. TransDecoder (v5.5.0) was used to annotate coding regions within transcripts. Blastp (v2.10.1, default parameters) was used to alignment

the rye B drive candidates to the translated transcriptome of *Ae. speltoides* + B.

## Phylogenetic analysis

The protein sequence of DCR28 was aligned to PANTHER and used in a BLASTp search in the NCBI non-redundant (nr) protein sequence database to identify DCR28-like proteins. We obtained 434 sequences from 325 species which covered between 25% and 100% of the query length. The sequences were aligned by ClustalX[95]. After removing very similar sequences or sequences lacking conserved modules, we restricted the dataset to a conserved region of 80 amino acids length where the alignment is unambiguous and used 139 sequences covering the major branches of plants from algae to angiosperms. Phylogenetic analyses were conducted through heuristic searches in PAUP* 4.0a169[96] using the maximum-likelihood algorithm with JTT rate matrix, empirical state frequencies and molecular clock not enforced. A smaller dataset consisting of a subset of 39 representative sequences was used to illustrate the results of the phylogenetic analyses i.e. identify the position of B-specific *DCR28* within the gene family, using the same settings as before.

## Cloning of reporter constructs

The EYFP and mCherry fluorescent tags were fused to the gene of interest using the Golden Gate assembly method[97]. Phusion High-Fidelity DNA Polymerase (New England Biolabs, NEB) was used for PCR following the protocol recommended by NEB. DCR28 was amplified from cDNA of anther of wheat + 2B$^s$ undergoing the first pollen mitosis using primers spanning both 3′ and 5′ untranslated regions (UTRs). The primers (DCR28_5_UTR, DCR28_3_UTR; Supplementary Table 4) were designed as described above. The PCR products were purified using the Monarch PCR & DNA Cleanup Kit (NEB). Subsequently, a second round of PCR was performed using a pair of primers (NT-DCR28F and NT-DCR28R), which included overhangs containing a *Bsa*I site, to amplify the purified product from the start codon to the stop codon of DCR28. The second PCR product was purified and used for Golden Gate assembly.

The coding sequence of DCR28-like rye was obtained from two combined DNA fragments. Fragment DCR28A2 (primer: EYFP28A2F and EYFP28A2R, Supplementary Table 4) was amplified and purified using cDNA of the rye root meristem as a template. PCR amplification of the fragment DCR28A1 failed due to its high GC content. Therefore, DCR28A1 was codon optimized (https://eu.idtdna.com/pages/tools/codon-optimization-tool) and synthesized by Eurofins Genomics GeneStrands (Supplementary Data 11). For Golden Gate assembly, the reaction contained 1.5 μL T4 ligase buffer, 0.5 μL T4 DNA ligase, and 1 μL *Eco*31I (FastDigest) from Thermo Fisher Scientific. The assembly included 100 ng of each component: destination vector PICH86966, pICH41295:35Sprom (35S promotor), pICH41258-mCherry-6xGly or pICH41258-EYFP-6xGly, purified PCR product(s), pAGM9121-rbcSE9ter (terminator) (Supplementary Table 6). Finally, ddH$_2$O was used to fill up the reaction to 15 μL. The reaction was mixed and cycled between 37 °C (3 min) and 16 °C (5 min) for 20 cycles, and then subjected to a final inactivation for 5 min at 80 °C and kept at 10 °C. Fifteen microliters of the assembled mixture was transformed into the *E. coli* TOP10 competent cells by heat shock. Sanger sequencing was used to verify the accuracy of the vectors.

## Transient protein expression in *N. benthamiana*

The cell-cycle dynamic of *DCR28* was characterized in a *N. benthamiana* cell division-enabled leaf system[48]. *N. benthamiana* with transgenic CFP-histone H2B was used to visualize the dynamics of the chromosome and determine the stages of mitosis. First, each vector was transformed into Agrobacteria GV3101 separately and grown in a YEB liquid medium (5 g/L beef extract,1 g/L yeast extract, 5 g/L peptone, 5 g/L sucrose, 0.5 g/L MgCl$_2$) supplemented with rifampicin

(50 ng/L), gentamycin (30 ng/L), and vector-specific antibiotics. After 48 h incubation, the agrobacteria were collected by centrifugation (1500 × *g* for 20 min) and resuspended in the infiltration buffer (10 mM MES (2-morpholinoethanesulphonic acid, pH 5.6) supplemented with 10 mM MgCl$_2$). The spectrophotometer was used to measure the concentration of each suspension using 600 nm (OD$_{600}$). For co-infiltration, each agrobacterial suspension was diluted to a final OD600 of 0.4 except CycD3 (Supplementary Table 6), which was 0.6. Afterwards, they were mixed in equal volumes and incubated at RT for 3–24 h. p19 (Supplementary Table 6) was used to suppress post-transcriptional gene silencing in all the mixtures. For the mixtures which were used to study cell division, CycD3 is required[48]. To inject the agrobacterial suspension into the leaves of 4-week-old plants, a blunt-tipped Soft-Ject® syringe was used. After infiltration, the plants were further grown in a 25 °C growth chamber for 36–48 h. A confocal laser scanning microscope (LSM 780, Carl Zeiss GmbH) was used for microscopic analysis. Images were taken with a 40× (C-Apochromat 40x/1.20 W Korr FCS M27) water immersion objective and the 'ZEN 2012 SP5 FP3' software version 14.0 (Carl Zeiss GmbH). Raw images were processed with Fiji version 2.3.051. Multichannel images and stacks were split by color, and their brightness and contrast were adjusted individually for each channel. Stacks were merged into one image by maximum intensity projection if a single image from the stack was not representative. A 10 μm scale bar was added to the images of the different channels before they were saved individually and merged into one image.

## Reporting summary

Further information on research design is available in the Nature Portfolio Reporting Summary linked to this article.

## Data availability

Raw sequence reads are available from the European Nucleotide Archive (ENA) under accession PRJEB69479 (Supplementary Table 7) (whole-genome sequencing short reads, PacBio, Nanopore, Hi-C sequencing, optical mapping), and PRJEB46034 (RNA-seq). The GCA assembly ID of the B chromosome assembly is GCA_964027155 [https://www.ebi.ac.uk/ena/browser/view/GCA_964027155]. Source data are provided with this paper.

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

## Acknowledgements

We would like to thank T. Endo (Japan) and L. Morais (Portugal) for providing the valuable +B wheat and +B rye genotypes; M. Goodin (USA) for sharing a *N. benthamiana* reporter line; S. Somasundaram and A. Korte for experimental support; O. Weiss, K. Kumke, S. Swetik, S. König, M. Knauft, I. Walde for their excellent technical assistance; A. Fiebig and M. Maruschewski for sequence submission (IPK, Germany); P. Cápal, M. Said, and Z. Dubská for chromosome sorting; H. Toegelová (IEB Olomouc, CZ) for help in generating the optical map; A. Koblížková (Biology Centre, Ceske Budejovice, CZ) for extracting HMW DNA and the ELIXIR infrastructure (CZ projects ID:90255, ID:90254 supported by the

Ministry of Education, Youth and Sports, CZ) for providing computational resources. This work was supported by the China Scholarship Council (CSC202006850005) to J.C.; the DFG (HO1779/30-1 and 30-2; HO1779/34-1) to A.Ho.; the ELIXIR-CZ Research Infrastructure Project (LM2023055) to J.M.; the DFG (BU2955/1-2) to K.B., and by the project TowArds Next GENeration Crops, reg. no. CZ.02.01.01/00/22_008/0004581 of the ERDF Programme Johannes Amos Comenius to J.Ba., Z.T., and H.S.

## Author contributions

J.C. performed most of the experiments, including plant cultivation, FISH, genome assembly, transcriptome analysis, PCR, cloning, and transient expression; A.B.: material preparation and transcriptome analysis; A.V.: transient expression; F.R.B. and A.V.: phylogenetic analysis; V.S. performed super-resolution microscopy; J.F. conducted flow cytometry; A.Hi., T.S., and J.M. performed sequencing; Z.T. and H.S.: optical mapping; J.Ba., T.R.-W., and S.D.: data analysis; K.B. and J.B.: transient expression; G.K.: experiments in *Ae. speltoides*. A.Ho. supervised the research project; J.C., T.R.-W., and A.Ho. wrote the manuscript with the help of all coauthors.

## Funding

## Competing interests

The authors declare no competing interests.
