## [Peer Review file · Nature Communications]

The genetic mechanism of B chromosome drive in rye illuminated by chromosome-scale assembly

Corresponding Author: Professor Andreas Houben

Version 0:

Reviewer comments:

Reviewer #1

(Remarks to the Author)

This submission reports a high-quality sequence of the rye B chromosome. This chromosome has long been a model for understanding the drive mechanism of a supernumerary chromosome. Using variants that possess the drive mechanism or not in rye or introduced into wheat, candidates for genes involved in the drive mechanism were narrowed. Using rye B variants in wheat was a clever approach to help with the sequencing and assembly. The paper is very clearly written.

Line 31, maybe use “system” instead of “species”.

Line 61, delete the word “line”. There is no germline in plants; that is an animal phenomenon. Using “germinal cells” is fine.

Line 73-76, maybe this sentence should be edited as follows considering that it is mentioned in the Discussion: “...severely limited. A recent effort by Blavet et al. (2021) succeeded in producing a draft assembly of a maize B chromosome that narrowed the cis and trans sites involved with its drive mechanism.”

Line 208 & 217 & 233: Delete the word “by”.

Line 307: The DCR169 transcripts are not mentioned again. Is there an argument that would eliminate their candidacy for involvement in the drive?

Is there any unusual behavior of microtubules that have been observed with the B chromosome in rye (or in wheat with the rye B) at the first pollen mitosis? If DCR28 is indeed the best candidate, might one not think there would be?

Line 453: maybe “...of rye and maize B chromosome drive, none of the 34 candidates for a trans-acting, protein encoding drive gene of the maize....”

Lines 489-495: This should probably be stated better. The genes do not duplicate to prevent the extinction of the drive mechanism, but rather, when duplications occur, they must be selected for increased strength of the drive mechanism. Then, once duplicated, the drive controlling genes can be reborn.

Lines 497-505: The authors seem conflicted about whether DCR400 could be involved in the drive mechanism. It's predicted function might make it a good candidate but then it is argued that it is a “young” paralog, which would seem to disqualify it. But then they argue that the drive is unlikely to be regulated by a single gene (but it could, of course, be regulated by a single gene). A test could come from transforming it into wheat and seeing if it salvages the drive mechanism of the drive negative B's that were used. A salvage would implicate it; a negative result might mean it is not involved or needs another component.

Lines 557-587: If the B has paralogs on all A chromosome that originated from chromoanagenesis, what is left in the nucleus to recover the neo B? One might argue that the other chromosomes were reassembled, but this seems unlikely given the excellent synteny of rye with its relatives. With the maize B, there are also paralogs from across the genome, but their divergence times are very different among them. With the proposed scenario for the rye B, the divergence with A paralogs would cluster. Do they? The maize result was rationalized as the result of transposition of genes from different A

chromosomes over evolutionary time. These two alternatives could easily be distinguished by checking the divergence times between the B and A paralogs. This should be done.

Based on tissue expression, B specific expression, and an effect on microtubules in infiltrations of *N. benthamiana*, a reasonable candidate for the drive gene seems to be the DCR28 gene array. Unfortunately, we don't know what effect this gene might have in rye or wheat with B chromosomes. The authors raise a little doubt in suggesting that DCR400 might be involved. Transforming wheat with DCR28 and testing for rescue of the drive mechanism of the drive-negative B's introduced into wheat might have been a nice validation. But, as noted, the argument that DCR28 is the drive gene is reasonable.

Reviewer #2

(Remarks to the Author)

Comments:

The B chromosome of rye undergoes targeted nondisjunction during pollen mitosis, this paper studies the mechanism of B chromosome drive in rye by assembling the rye B chromosome and identifying five candidate genes as trans-acting moderators of chromosome drive through comparative RNA-seq between rye B chromosome variants. The reported results are both interesting and significant, as they address for the first time the longstanding question of B chromosome nondisjunction in plants. The assembled genome and the discovered drive mechanism provide a crucial foundation and reference for studying B chromosomes in rye and other plants. Moreover, the paper is clearly written and easy to follow. I have only some points for consideration by the authors:

1. When assessing the quality of genome assembly, additional information should be provided to explain how many gaps are present in the assemblies, particularly in the B chromosome. Furthermore, the evaluation of genome assembly continuity should include the annotation of long terminal repeats (LTR) assembly Index (LAT) score.
2. In the Hi-C interaction heatmap of the wheat scaffolds with the rye B chromosomes, the Hi-C successfully scaffolds the 21 wheat chromosomes but fails to scaffold the rye B chromosome, what is the quality of the assembled B chromosome?
3. Line 243-245: "Approximately 36 Gb (~2x coverage) of short-read genome sequencing data were mapped to the B pseudomolecule and unplaced B contigs...." How can unplaced contigs be distinguished between wheat chromosomes and the B chromosome?
4. The DCR-located drive candidate genes were annotated used the PANTHER, could gene knockout experiments be conducted to further validate whether the identified candidate genes regulate B chromosome drive?
5. Do homologous genes for these candidate genes also exist on the B chromosomes of other species?
6. The authors identified five candidate genes that control rye B chromosome drive. I would be interested to hear the authors' thoughts on whether B chromosome drive is controlled by a single gene or regulated by multiple genes. Are there other factor may involved in this regulation, and if these genes were transferred to other chromosomes, would they induce nondisjunction?

Minor comments:

In Fig. 4, bold the text in (c).

Line 78: The ~560 Mb... (instead of Mbp)

Reviewer #3

(Remarks to the Author)

General comment

The manuscript entitled "The genetic mechanism of B chromosome drive in rye illuminated by chromosome-scale assembly", by Chen et al., addresses the main question about B chromosomes in one of the best known systems, that of rye, reaching hitherto almost unknown depth about the molecular mechanisms by which these genomic parasites manage to establish and maintain themselves in populations. The presence of functional genes for ribosomal DNA was long ago inferred from nucleolus formation by grasshopper B chromosomes (Bidau 1986; López-León et al. 1991). The first protein genes in B chromosomes were reported in the migratory locust, but they were also for repetitive DNA, specifically coding for the H3 and H4 histone genes, and their possible functionality was unknown (Teruel et al. 2010). However, the presence of functional protein genes on B chromosomes was ignored for long, simply because their search was technically unapproachable for chromosomes whose DNA is mostly repetitive. Now, thanks to recent improvements in massive sequencing techniques, yielding longer and longer reads, the authors of this manuscript have reached a major milestone in assembling a ~430 Mb large pseudomolecule representing ~77% of the rye B. Until now, the closest thing to this was a 297 Mb assembly in 444 scaffolds for the B chromosome of *Nasonia vitripennis* (Benetta et al. 2020), and 125.9 Mb of for the B chromosome of maize, representing 89% of its predicted 141-Mb size (Blavet et al. 2021). Therefore, the work carried out in this manuscript is very meritorious, as rye genome is thrice that of maize, and 23 times larger than that of *Nasonia*. In fact, it will be very difficult to perform a work like this in non-model organisms lacking the technical facilities existing in rye.

But even more important is the finding of a couple of genes functionally involved in B drive. In *Nasonia*, a gene involved in drive (named haploidizer) was also found, but this was of bacterial origin and was not found in the A genome of the species. In maize, no functional genes could be ascribed as responsible for B drive (Blavet et al. 2021). In rye, however, the two genes found (DCR28 and DCR400) are also part of the A genome of rye and, at last, we are witnessing how the drive of a B chromosome is possible thanks to the involvement of genes derived from the host genome, an unprecedented knowledge until now, thanks to the present manuscript.

The methodological approach has been the basis of the success of the achievements of this research. Having the possibility of sequencing the B chromosome in the wheat genome, instead of the host (rye) genome, has avoided interference from sequence homology between the B and A genes of rye. Also carrying out the functional analysis of the candidate genes responsible for B drive is an additional merit which logically increases the quality of this manuscript even more. But I want to highlight that this work has been possible thanks to the previous work by some Newton's giants (e.g. Neil Jones in UK and María Puertas in Spain), who laid the foundations over many years to make it possible to get here, and who, in my opinion, should be given more credit in this historical paper on rye B chromosomes. Of course, it had to be Andreas Houben's group that achieved this milestone, which rounds off his career of technical excellence since I met him for the first time, in 1993, during the first B chromosome conference, where he presented the chromosome microdissection technique, which many of us have done later when he was already immersed in other innovations.

That said, I have no doubt that this work should be published and, with the aim of contributing as much as possible to improving the presentation, so that a work as complex and dense as this one is accessible to a general audience, I make the following minor comments.

Minor comments:

Lines 53-61. I think that this paragraph explaining the evolutionary arms race between B chromosome and their host genome should be revised, as, to my understanding, some of the cause-consequence descriptions are presented backwards. Parasites in general are not considered parasites because they cause abnormal phenotypes, stress, disease or reduced fertility. Rather, they are considered parasites because they exploit host mechanisms to ensure their own transmission and maintenance within populations. Stress, disease or reduced fertility are byproducts (side-effects) of parasitism. Likewise, B chromosomes are considered parasitic because they have mechanisms (chromosome drive, in this case) that allows them to use the cellular machinery to achieve their own transmission, even if they are neutral or detrimental to the host. Drive is also not only a mechanism to by-pass suppressors. If the chromosome was not driving before the appearance of suppressors, suppressors would not arise, but the chromosome would be eliminated by drift in a few generations. Therefore, the evolutionary arms race model should begin with chromosome drive, then selection favouring the evolution of suppression, then stronger or different drive mechanisms, then stronger or different suppressors, etc. Arms race description in these lines highly fits the near-neutral model devised by Camacho et al. (1997, 2000) and Camacho (2005). González-Sánchez et al. (2003) applied this model to the case of maize. In my opinion, the near-neutral model also applies to B chromosomes in rye as the A-B chromosomes arms race explains why B-drive and A suppressor genes could have had successive turnovers across evolution, according to authors' discussion, and new genes could have been recruited to the B chromosome showing the "regeneration" stage of the model by which one B variant is replaced by another displaying a more efficient drive mechanism. As this manuscript demonstrates, this may be based on a new B-gene providing more efficient drive in new environmental circumstances, or else on a newly recruited gene coming from the A genome which provides this same advantage for the B in terms of its drive.

Line 65: "Drive mechanisms in B chromosome systems have been studied in many species" How many, approximately? Can the authors provide some proportions and references?

Lines 137: "A previously-undocumented deficient B variant named Bk-3 was identified when we screened the progeny of wheat with Bk-2" Do the authors mean that this variant arose de novo among the progeny of wheat with Bk-2?

Figure 3c and 3d: It would be helpful to include the approximate location of the drive control region in Figure 3c as well, for clarity.

Lines 246, Suppl. Table 4: Header of column A is missing. Do asterisks mean something in column H? Information in columns I and J (primary and secondary confidence class) are not explained in the manuscript. Also, to facilitate reading, it would be better that each supplemental file contains its table caption.

Line 259. Figure 4a is a bit hard to understand, please consider redesigning, or including more explanations in the legend.

Lines 271: "Four genes (DCR398, DCR399, DCR145, and DCR154) appear to be located not only within the DCR but also at other positions on the rye B chromosome" Since this conclusion is based on PCR results, and there is no OB control, I wonder if it is possible that these PCR amplifications might be due to primers inespecifically targeting homologous sequences from genes located somewhere in the A chromosomes of the wheat genome. However, in the methods section they mention "B chromosome-specific primers", which may indicate that these primers were designed to amplify only the B chromosome version. Please, clarify this point.

Lines 294: "The Futsch-like family is unknown in plants" Do the authors mean the function is not described in plants? In a quick database search in Uniprot, I found several entries of "microtubule associated futsch-like protein" in *Arabidopsis thaliana*, for instance, so the protein is known in plants; although papers describing its function are all in animals.

Line 300: "DCR400 not only shows stronger expression in shoot apical meristems" Figure 6a shows differential expression of DCR28 and DCR400 in both A and B chromosomes. Perhaps I have not been able to find in the manuscript how A and B derived transcripts were identified. In case of DCR28, perhaps transcript length worked but, in DCR400, lines 352-354 indicate that "DCR400 shows exceptionally high amino acid identity to the rye paralog gene". In case of B-carrying rye plants, both A- and B-derived transcripts need to be identified to build a Figure like 6a. If it is not described in the manuscript, please do so. In case of DCR400, Figure 6a shows that DCR400 expression was much higher in 2B than in 1defB and 0B, but the TPM is roughly 50/50 distributed between DCR400 and DCR400-like rye. I wonder if this might be caused by the mapping algorithm not being able to distinguish between DCR400 and DCR400-like rye, and thus it mapped roughly half the reads to each of them in most of their length. Even so, the amount of DCR400 transcripts was higher in 2B than 0B and 1defB suggesting the differential expression of this gene in 2B plants. Please clarify this point.

Line 336: "Local gene clusters are suggestive of duplication events mediated by local microhomologies via (e.g.) repeat arrays that lead to strand slippage during replication" There has been several instances of meiotic drivers that have been amplified via tandem duplication (eg. *Slx1* and *Sly* in mouse; Cocquet et al. 2009; Kruger et al. 2019) or somehow spread in association with other repetitive DNA or transposable sequences, such as *wtf* spore killers in *S. pombe* (Eickbush et al. 2019) and *Dox* in *Drosophila* (Muirhead et al. 2021). Yet the mechanism by which they amplified is still not demonstrated. I suggest to cite these papers here, making clear that the slippage is hypothetical, also discussing some alternative hypotheses, if applicable (eg. ectopic recombination, nonallelic gene conversion...).

Lines 350-351: "though DCR28 encodes a protein 71 residues shorter than its A chromosome counterpart" Maybe I missed it, but it is not clear to me how the rye B transcripts are assembled, neither how the author assign them to the B chromosome. Please provide more details on this, and also make clear that this is a predicted protein based on nucleotide sequence from those transcripts.

Line 354: "suggesting a more recent copying event" Or, alternatively, the existence of high functional constraints for B drive.

Line 517: Change "has also be reported" for "has also been reported"

Lines 577-578: "To counteract purifying selection, the gain of a chromosome drive mechanism is a prerequisite for the transgenerational stability of a newly formed B" As beneficial effects of the newly formed proto-B are unlikely, the safest pathway by which it can remain in a population is by displaying drive from the beginning. This initial drive does not necessarily has to be the same observed for current rye Bs. The existence of many gene copies of the driver gene, with some sequence differences between them, would support that drive mechanism could have changed across evolution, probably due to an arms race dynamics. In the beginning, drive could even occur prior or during meiosis, and later the appearance of other drive mechanisms might mask the ancestral mechanisms. These comments complement those about Lines 53-61, as the extraordinary molecular details revealed in the present manuscript indicate that the B chromosome system in rye clearly fits the near-neutral model of B chromosome evolution.

References

- Bidau, C. J. (1986). A nucleolar-organizing B chromosome showing segregation–distortion in the grasshopper *Dichroplus pratensis* (Melanoplinae, Acrididae). *Canadian Journal of Genetics and Cytology*, 28(1), 138–148. <https://doi.org/10.1139/g86-018>
- Benetta, E. D., Antoshechkin, I., Yang, T., Nguyen, H. Q. M., Ferree, P. M., & Akbari, O. S. (2020). Genome elimination mediated by gene expression from a selfish chromosome. *Science Advances*, 6(14), eaaz9808. <https://doi.org/10.1126/sciadv.aaz9808>
- Blavet, N., Yang, H., Su, H., Solanský, P., Douglas, R. N., Karafiátová, M., Šimková, L., Zhang, J., Liu, Y., Hou, J., Shi, X., Chen, C., El-Walid, M., McCaw, M. E., Albert, P. S., Gao, Z., Zhao, C., Ben-Zvi, G., Glick, L., ... Bartoš, J. (2021). Sequence of the supernumerary B chromosome of maize provides insight into its drive mechanism and evolution. *Proceedings of the National Academy of Sciences of the United States of America*, 118(23), 1–11. <https://doi.org/10.1073/PNAS.2104254118>
- Camacho, J. P. M., Shaw, M. W., López-León, M. D., Pardo, M. C., & Cabrero, J. (1997). Population dynamics of a selfish B chromosome neutralized by the standard genome in the grasshopper *Eyprepocnemis plorans*. *The American Naturalist*, 149(6), 1030-1050.
- Camacho, J. P. M., Sharbel, T. F., & Beukeboom, L. W. (2000). B-chromosome evolution. *Philosophical Transactions of the Royal Society of London. Series B: Biological Sciences*, 355(1394), 163-178.
- Camacho, J. P. M. (2005). B chromosomes. In *The evolution of the genome* (pp. 223-286). Academic Press.
- Cocquet, J., Ellis, P. J., Yamauchi, Y., Mahadevaiah, S. K., Affara, N. A., Ward, M. A., & Burgoyne, P. S. (2009). The multicopy gene *Sly* represses the sex chromosomes in the male mouse germline after meiosis. *PLoS biology*, 7(11), e1000244.
- Eickbush, M. T., Young, J. M., & Zanders, S. E. (2019). Killer meiotic drive and dynamic evolution of the *wtf* gene family. *Molecular Biology and Evolution*, 36(6), 1201-1214.
- González-Sánchez, M., González-González, E., Molina, F., Chiavarino, A. M., Rosato, M., & Puertas, M. J. (2003). One gene determines maize B chromosome accumulation by preferential fertilisation; another gene (s) determines their meiotic loss. *Heredity*, 90(2), 122-129.
- Kruger, A. N., Brogley, M. A., Huizinga, J. L., Kidd, J. M., de Rooij, D. G., Hu, Y. C., & Mueller, J. L. (2019). A neofunctionalized X-linked ampliconic gene family is essential for male fertility and equal sex ratio in mice. *Current Biology*, 29(21), 3699-3706.
- López-León, M., Cabrero, J., & Camacho, J. (1991). A nucleolus organizer region in a B chromosome inactivated by DNA

methylation. *Chromosoma*, 100(2), 134–138. <http://www.springerlink.com/index/U10420564J560M64.pdf>
Muirhead, C. A., & Presgraves, D. C. (2021). Satellite DNA-mediated diversification of a sex-ratio meiotic drive gene family in *Drosophila*. *Nature Ecology & Evolution*, 5(12), 1604-1612.
Teruel, M., Cabrero, J., Perfectti, F., & Camacho, J. P. M. (2010). B chromosome ancestry revealed by histone genes in the migratory locust. *Chromosoma*, 119(2), 217–225. <https://doi.org/10.1007/s00412-009-0251-3>

Reviewer #4

(Remarks to the Author)

I co-reviewed this manuscript with one of the reviewers who provided the listed reports. This is part of the Nature Communications initiative to facilitate training in peer review and to provide appropriate recognition for Early Career Researchers who co-review manuscripts

Version 1:

Reviewer comments:

Reviewer #1

(Remarks to the Author)

The presentation is much improved. The evidence that the authors have identified a trans-acting factor for the rye B is reasonable. Some suggestions on the current version follow.

Line 80, need a space between “although” and “a”

Line 82, best to delete the word “possibly”. The cis site for the maize B was well defined years ago (Han et al 2007, *Plant Cell* 19: 524-533) and a trans-acting factor even earlier (Birchler and Auger, 2002, *J. Hered.* 93: 42-47). The cited paper is fine because therein they were placed on the currently available maize B sequence.

Line 97, Delete “(germline)”

Line 107, Insert “the” between “at” and “first”

Line 109, Insert “a” between “of” and “standard”

Line 243, Remove extra space

Line 315, Delete “s” from “plays”

Line 415, Delete extra “that”

Line 419, Delete “of the genes”

Line 484, Insert “the” between “of” and “rye”

Lines 590 to 623, The authors suggest a scenario that chromosomes are missegregated and land in micronuclei in gamete development and undergo chromoanagenesis, rearrange, and organize again as a B chromosome. But then later, the authors wind up claiming that all 7 chromosomes get involved somewhat sequentially. Looking at Supplemental Figure 17, it does seem that chromosomes 6 and 7 were early contributors. The position on the B chromosome is given but how does a synteny relationship look like between those 6 and 7 paralogs on the B versus 6 and 7 paralogs on the A chromosomes, respectively? If there is good synteny between the B and A paralogs on 6 and 7, then rearrangement seems unlikely. The claim is that more chromoanagenesis occurred for the other five chromosomes in subsequent events. But why would the pieces insert into the B chromosome specifically rather than being splattered all over the genome? Rye and barley have good synteny, so that seems unlikely. I guess one could argue that such fragments landing in the A chromosomes would be selected against due to aneuploidy, but the same would apply to the B chromosome unless the genes are silenced, which the RNA data suggests otherwise. One can see that as one builds on this scenario, it might be an “air castle”. One solution would be to delete this whole section and Suppl Figure 17. Further work might eventually lead to an understanding that seems reasonable.

Reviewer #2

(Remarks to the Author)

After reading the revised version and response letter, I think the authors provide thoughtful replies and corresponding revisions to all of my comments. I have no further comments.

Reviewer #3

(Remarks to the Author)

Authors have adequately addressed my concerns and suggestions. However, I still have a few remarks, the most important concerning the analysis requested by Reviewer 1 about sequence identity between rye A and B chromosome paralogs (Suppl. Figure 17). In my opinion, BLASTN might not be the best tool for this analysis, as it throws multiple hits for every single region of the B chromosome; and the parameters used for this search are likely too relaxed (Section 17 of the Methods) and could lead to inaccurate results. I suggest performing this analysis with more stringent BLAST search parameters and limiting the search to the best hit (or else explaining the rationale for the used parameters). However, a better approach would be to perform whole genome alignment of each A chromosome to the B pseudomolecule (masking repeats, and using LASTZ, for example), or extracting the exon and intron sequences from the A chromosomes annotation (with bedtools) and map them to the rye B pseudomolecule (using minimap2, GMAP or STAR for instance). Then, the authors should be able to calculate a gene-based or window-based divergence between A and B genes. Also, Suppl. Figure 17 contains mixed information about introns and exons, which is tricky because the models for molecular evolution of introns and exons are quite different. Most substitutions in introns are expected to be neutral, therefore the number of substitutions in introns (molecular divergence) is expected to be somewhat proportional to time. Substitutions in exons, however, may be constrained by selection. Additionally, introns may contain transposable elements (TEs) or other types of repeats, which could generate noise in inferring the A chromosome gene contribution to the B chromosome. For this task to be reliable, it is absolutely necessary to avoid the presence of repetitive DNA that may be present on two or more A chromosomes. Authors discarded genes annotated as TEs, but they did not mask repeats in the assemblies (at least I have not found information about this in the revised manuscript). For instance, the presence of other kinds of repeats, such as tandem repeats or TE fragments, inserted within the introns of the genic sequences, might give misleading results, since they could be present in several A chromosomes. Even masking known TEs, it is still conceivable the presence of a few still undescribed TEs in the rye genome which could not be masked. These repeats or repeat fragments could explain why all the length of the B chromosome pseudomolecule shows homology with most A chromosomes, as shown in Suppl. Figure 17. For this reason, I believe that restricting the analysis to exonic sequences would give a clearer picture of B chromosome origin. In any case, and given that I did not suggest this analysis, as far as I am concerned, it is not essential to maintain this new section and the authors could eliminate it from the manuscript if they consider it convenient.

Minor comments.

- Authors provide the captions for Supplementary tables in the main text, but these captions are not included as headers within the tables themselves (xls files). In my opinion, the captions should be added to the xls files to facilitate reader comprehension. In the current version, the names of the Supplementary tables are not even shown in the file names, so readers must guess which of the nine supplementary tables they are looking at based on the information in the table.
- Figure S12b might include a typo in sequence names (NCR400 instead of DCR400).
- Figure S17 is presented in the Discussion rather than the Results section, which is fine. However, if it is kept in the final version, some information is missing. For subfigure A: Modify the X-axis label to clearly indicate that it represents sequence identity with B-chromosomal sequences. The term "Frequency" typically refers to a proportion, but the color legend extends beyond 600. Please clarify whether this number represents the number of genes or something else (as per Section 17 of the Methods, I assume it might be the number of BLAST hits, which would indeed be higher). For subfigure B: please make it clear in the legend whether each dot represents a gene or a BLAST hit.

Reviewer #4

(Remarks to the Author)

Version 2:

Reviewer comments:

Reviewer #1

(Remarks to the Author)

The authors were wise in their decisions. I have no further comments and look forward to the publication.

Reviewer #3

(Remarks to the Author)

All my suggestions have been addressed and the manuscript has very high quality and deserves publication.

Reviewer #4

(Remarks to the Author)

Dear Reviewers,

We would like to thank you for providing a thorough, constructive, and overall positive review of our manuscript. We are confident to have addressed most comments and concerns. This led to a substantially revised version of our manuscript.

Below, we provide details in a point-by-point response to all remarks and questions. We hope that our manuscript now meets the high-quality standards of Nature Communication.

With best regards on behalf of all authors,

Andreas Houben

REVIEWER COMMENTS

Reviewer #1 (Remarks to the Author):

This submission reports a high-quality sequence of the rye B chromosome. This chromosome has long been a model for understanding the drive mechanism of a supernumerary chromosome. Using variants that possess the drive mechanism or not in rye or introduced into wheat, candidates for genes involved in the drive mechanism were narrowed. Using rye B variants in wheat was a clever approach to help with the sequencing and assembly. The paper is very clearly written.

Line 31, maybe use “system” instead of “species”.

RESPONSE: Many thanks and we took this suggestion.

Line 61, delete the word “line”. There is no germline in plants; that is an animal phenomenon. Using “germinal cells” is fine.

RESPONSE: Many thanks and we took this suggestion.

Line 73-76, maybe this sentence should be edited as follows considering that it is mentioned in the Discussion: “...severely limited. A recent effort by Blavet et al. (2021) succeeded in producing a draft assembly of a maize B chromosome that narrowed the cis and trans sites involved with its drive mechanism.”

RESPONSE: Many thanks and we took this suggestion.

Line 208 & 217 & 233: Delete the word “by”.

RESPONSE: Many thanks and we took this suggestion.

Line 307: The DCR169 transcripts are not mentioned again. Is there an argument that would eliminate their candidacy for involvement in the drive?

RESPONSE: We reconsidered our assumption and toned down our conclusion about the unlikely function of *DCR169* in the process of drive. Compared to *DCR28*, *DCR169* is not only active in tissue where nondisjunction happens (Figure 4b). However, interestingly, all five transcript variants of *DCR169* share similarities with long non-coding RNA-producing B-specific satellite E3900 (Supplementary Fig. 8) and 3 of them may encode an ~160 aa unknown protein (Supplementary Table 7). The probable *gag* gene origin of E3900 from a centromere-specific retrotransposon element (Langdon et al. 2000) lends some support that E3900 may play a not-yet-defined role in the process of nondisjunction in combination with additional controlling elements.

Now it reads: "All 5 transcript variants of *DCR169* share similarities with long non-coding RNA-producing B-specific satellite E3900 (Supplementary Fig. 8) and 3 of them may encode an ~160 aa unknown protein (Supplementary Table 7). However, the probable *gag* gene origin of E3900 from a centromere-specific retrotransposon element (Langdon et al. 2000) lends some support that E3900 may play a not-yet-defined role in the process of nondisjunction in combination with additional controlling elements".

Is there any unusual behavior of microtubules that have been observed with the B chromosome in rye (or in wheat with the rye B) at the first pollen mitosis? If DCR28 is indeed the best candidate, might one not think there would be?

RESPONSE: We agree with your assumption. In previous experiments we immunostained pollen (at first pollen mitosis) sections of +B plants with an alpha-tubulin antibody. We observed, an asymmetric cell division as expected and lagging Bs (Banaei-Moghaddam et al. 2012). We could not observe an obvious unusual behavior of microtubules in +B pollen. However, 0Bpollen was not analysed as control. Later we intended the generation of a DCR28-specific antibody. But we failed, because our peptide-based antibody did not produce signals after indirect immunostaining.

To overcome this problem, we now aim to generate a wheat (genotype: Fielder) DCR28-GFP reporter line (in combination with a chromatin reporter, like histone H2A-tdTomato), which after crossing with +B wheat will be used for pollen life imaging. Unfortunately, this live imaging

experiment requires at least one year (8 months for wheat transformation and additional months for crossing with different +B genotypes).

Line 453: maybe "...of rye and maize B chromosome drive, none of the 34 candidates for a trans-acting, protein encoding drive gene of the maize..."

RESPONSE: Many thanks and we took this suggestion.

Lines 489-495: This should probably be stated better. The genes do not duplicate to prevent the extinction of the drive mechanism, but rather, when duplications occur, they must be selected for increased strength of the drive mechanism. Then, once duplicated, the drive controlling genes can be reborn.

RESPONSE: Many thanks for the suggestion: Now it reads: "De Carvalho et al. (2022) proposed that when duplications of drive-controlling genes occur, they must be selected for increased strength of the drive mechanism. Then, once duplicated, the drive-controlling genes can be reborn (De Carvalho et al. 2022)."

Lines 497-505: The authors seem conflicted about whether DCR400 could be involved in the drive mechanism. It's predicted function might make it a good candidate but then it is argued that it is a "young" paralog, which would seem to disqualify it. But then they argue that the drive is unlikely to be regulated by a single gene (but it could, of course, be regulated by a single gene). A test could come from transforming it into wheat and seeing if it salvages the drive mechanism of the drive negative B's that were used. A salvage would implicate it; a negative result might mean it is not involved or needs another component.

RESPONSE: Yes, *DCR400* is tricky and we added following. "It is conceivable that the gene composition needed for drive evolved successively, with *DCR400* emerging as a component that adds to drive of *the* rye B chromosome in more recent evolution."

The proposed transformation experiment of wheat possessing a drive negative Bs is on our research agenda. However, as explained above, this experiment takes at least one year.

Lines 557-587: If the B has paralogs on all A chromosome that originated from chromoanagenesis, what is left in the nucleus to recover the neo B? One might argue that the other chromosomes were reassembled, but this seems unlikely given the excellent synteny of rye with its relatives. With the maize B, there are also paralogs from across the genome, but their divergence times are very different among them. With the proposed scenario for the rye B, the divergence with A paralogs would cluster. Do they? The maize result was rationalized as the result of transposition of genes from different A chromosomes over evolutionary time. These two alternatives could easily be distinguished by checking the divergence times between the B and A paralogs. **This should be done.**

RESPONSE:

To address this question, we performed an additional analysis in which we tested for sequence identity between genomic sequences, including exons and introns, of A chromosomal genes, and sequences on the B chromosome pseudomolecule. In this context, we interpret sequence identity as a proxy for divergence time between A and B paralogs, with older sequences showing lower sequence identity (similar to the approach used by {Martis, 2012 #14464}). This analysis showed that gene sequences of all A chromosomes cluster at certain divergence times. Notably, it seems that A chromosomes 6 and 7 contributed first, followed by 4, then followed by 1, 2, 3, and 5 to the formation of the former proto-B chromosome. This suggests that A chromosomes 6 and 7 contributed first to the origin of the proto-B through a chromoanagenesis-like event. Subsequently, the B chromosome accumulated further sequences from the entire A chromosome complement over evolutionary time, which can be seen by higher sequence identities between A and B paralogs. This is now included as Supplementary Figure 17. The main text (Materials and Methods, Discussion) was changed accordingly.

In Material & Methods

Sequence similarity between A and B paralogs

Genomic sequences, including exons and introns, of high-confidence genes located on A-chromosomes (Lo7 genome assembly v1) were used as query in a BLASTN search using the B chromosome pseudomolecule as reference. The BLASTN output was filtered for e-value $\leq 1 \times 10^{-20}$, alignment length of a minimum of 50% the A-paralog, and genes annotated as transposons were removed.

In Discussion: Indeed, our sequence identity analysis between genomic sequences, including exons and introns of A chromosomal genes, and sequences on the B chromosome pseudomolecule, showed that gene sequences of all A chromosomes cluster at certain divergence times (Supplementary Fig. 17). We interpreted sequence identity as a proxy for divergence time between A and B paralogs, with older sequences showing lower sequence identity. Notably, it seems that A chromosomes 6 and 7 contributed first, followed by 4, then followed by 1, 2, 3, and 5 to the formation of the former proto-B chromosome. This suggests that A chromosomes 6 and 7 contributed first to the origin of the proto-B through a chromoanagenesis-like event. Subsequently after its independence, the B chromosome accumulated through time further sequences from the entire A chromosome complement, which can be seen by the differing sequence identities between A and B paralogs.

Based on tissue expression, B-specific expression, and an effect on microtubules in infiltrations of *N. benthamiana*, a reasonable candidate for the drive gene seems to be the DCR28 gene array. Unfortunately, we don't know what effect this gene might have in rye or wheat with B chromosomes. The authors raise a little doubt in suggesting that DCR400 might be involved. Transforming wheat with DCR28 and testing for rescue of the drive mechanism of the drive-negative B's introduced into wheat might have been a nice validation. But, as noted, the argument that DCR28 is the drive gene is reasonable.

RESPONSE: The proposed "drive rescue" experiment of wheat possessing a drive negative Bs is on our research agenda. Unfortunately, as explained above, this transformation and crossing experiment takes at least one year.

However, to find additional support for whether DCR28 is a drive-controlling gene, we asked whether also the B of the rye-related species *Aegilops speltoides* encodes a first pollen mitosis-active DCR28-like gene. *Ae. speltoides* was selected because of the rye-similar B chromosome drive mechanism (Wu et al. 2019; Mendelson and Zohary 1972). Our transcriptome analysis identified a first pollen mitosis active, B-specific DCR28-like gene in *Ae. speltoides*. The same gene B-specific gene was found in four different accessions of *Ae. speltoides*. Thus, B-specificity of DCR28-like in related species and first pollen mitosis activity supports the likely drive controlling function of DCR28

We added the following: "In addition, we tested whether the B chromosome of the related species *Aegilops speltoides* also encodes a first pollen mitosis-active DCR28-like gene. *Ae. speltoides* was selected because of the rye B chromosome-like drive mechanism (Wu et al.

2019). BLAST search of the *de novo* assembled transcriptome of anthers undergoing PMI from *Ae. speltoides* with B chromosomes identified a DCR28-like gene (called AesDCR28). AesDCR28 is 290-aa long and has 60% (161/265) positive amino acids compared to DCR28 of the rye B (Suppl. Fig. 9). Like in rye, the B paralog of DCR28 was shorter than the corresponding A chromosome variant. Further, genomic PCR showed the presence of AesDCR28 on all B chromosomes from four different *Ae. speltoides* accessions (Fig. 4c). Thus, both B chromosomes of rye and *Ae. speltoides* encode DCR28-like genes.”

Reviewer #2 (Remarks to the Author):

Comments:

The B chromosome of rye undergoes targeted nondisjunction during pollen mitosis, this paper studies the mechanism of B chromosome drive in rye by assembling the rye B chromosome and identifying five candidate genes as trans-acting moderators of chromosome drive through comparative RNA-seq between rye B chromosome variants. The reported results are both interesting and significant, as they address for the first time the longstanding question of B chromosome nondisjunction in plants. The assembled genome and the discovered drive mechanism provide a crucial foundation and reference for studying B chromosomes in rye and other plants. Moreover, the paper is clearly written and easy to follow. I have only some points for consideration by the authors:

1. When assessing the quality of genome assembly, additional information should be provided to explain how many gaps are present in the assemblies, particularly in the B chromosome. Furthermore, the evaluation of genome assembly continuity should include the annotation of long terminal repeats (LTR) assembly Index (LAT) score.

RESPONSE: Many thanks for this suggestion. In total, there are 38 gaps in the B chromosome pseudomolecule (Supplementary Table 2: 3. Final agp-Coloum E). We calculated the LTR assembly Index (LAI) (Ou et al. 2018) of the assembled rye B chromosome, and its LAI score was 19.19, which is close to gold quality (LAI> 20). We have added both features to the revised manuscript. Now it reads: “On the other hand, evaluation of the B- pseudomolecule using the Long Terminal Repeat (LTR) Assembly Index (LAI) resulted in an LAI score of 19.19, which is close to gold quality (LAI> 20)(Ou et al. 2018).”

2. In the Hi-C interaction heatmap of the wheat scaffolds with the rye B chromosomes, the Hi-C successfully scaffolds the 21 wheat chromosomes but fails to scaffold the rye B chromosome, what is the quality of the assembled B chromosome?

RESPONSE: Thanks for this comment. Specialized chromosomes have been notoriously difficult to sequence and assemble because of their complex repeat structure. For example, more than half of the Y chromosome was missing in the version of the human genome until recently. Numerous efforts from many scientists allowed the completion of the Y chromosome sequence assembly (Rhie et al. 2023). A similar assembly challenge was reported for the germline-restricted chromosomes (GRCs) of songbirds, which sequence were highly fragmented and only accounted for 36 - 75% of the estimated sizes (Schlebusch et al. 2023). A more comparable example is the maize B chromosome, its pseudomolecule is 106.6 Mb covered 75.6% of its size (106.6 /141 Mb) and cumulative length (125.9 Mb) represents 89% of its size (Blavet et al. 2021).

Our rye B pseudomolecule has a size of ~430 Mb, representing ~77% of the actual rye B size (560 Mb). In total we assembled 160 rye B contigs, the largest contig is ~93 Mb, and N50 is ~60 Mb, and cumulative length (~452 Mb) represents 81% of the actual size (Table 2). Given that the most repetitive regions of a genome are always the most resistant to assembly and the exceptionally repetitive nature of the B chromosome, we are confident this represents an exceptionally accurate assembly of the unique regions. The high N50 value suggests that all but the most extreme repetitive regions were manageable for the assembler, and that the missing portion of the assembly is almost certainly in collapsed long repeats, that add very little to any functional analysis.

Unfortunately, BUSCO assessments could not be applied because of the B chromosome's dispensable nature. Your first suggestion of using the LTR assembly Index (LAI) to evaluate assembly continuity was very helpful. The LAI score of our assembled B chromosome is 19.19, which is close to gold quality (LAI> 20).

Finally, we feel satisfied that the arrangement of the B scaffolds, while not perfect, is a good representation of the approximate relative positions of most of the B genome. This owes to the high N50 (meaning most of the genome is amenable to a high number of Hi-C links), coupled with the general absence of off-diagonal signals in the Hi-C contact plots. While some such signals represent inevitable ambiguity often caused by repeated sequence, such signals are the exception and not the rule, indicating an assembly order apt to answer the kinds of broad

structural questions we address and correspondence between the localisation on the assembly of probe sequences with known physical positions in the FISH images offers proof at the gross scale of what we are confident is the case at the ~10+ megabase scale.

3. Line 243-245: “Approximately 36 Gb (~2x coverage) of short-read genome sequencing data were mapped to the B pseudomolecule and unplaced B contigs....” How can unplaced contigs be distinguished between wheat chromosomes and the B chromosome?

RESPONSE: Thanks for this comment. We distinguished the wheat chromosome sequences and the rye B chromosome sequences based on HiFi read coverage and evidence from Hi-C data. As we assembled wheat possessing six standard rye Bs copies, the HiFi read coverage levels per contig are different between of wheat A (~14x) and rye B (~44x) chromosomes. Compared to short-read coverage, which can easily be affected by transposable elements, the high-fidelity PacBio long reads (>15 Kb) can provide an accurate coverage estimate on the contigs. Moreover, Hi-C scaffolding resulted in 21 long scaffolds (>500 Mb) (Supplementary Fig. 3) and their alignments to the reference genome of wheat cv. Chinese Spring (IWGSC RefSeq v2.1 assembly) revealed their 1 to 1 syntenic correspondence to near-complete wheat chromosomes (Supplementary Fig. 4; Supplementary Table 1). Since as none of the wheat pseudomolecules were found to comprise any of the contigs ~ coverage > 23x (Fig. 3b; Table 2), contigs whose HiFi read coverage > 23x were assigned to the rye B. In total, ~452 Mb contigs were assigned to the rye B chromosome. ~430 Mb of them were placed onto B-pseudomolecule, but still ~22 Mb rye B-contigs remained unplaced.

4. The DCR-located drive candidate genes were annotated used the PANTHER, could gene knockout experiments be conducted to further validate whether the identified candidate genes regulate B chromosome drive?

RESPONSE: The proposed knockout experiment of candidate genes is on our research agenda. We aim to transform wheat (genotype: Fielder) with DCR-specific CRISPR constructs. After the successful generation of transgenic plants, we will perform crossing experiments with +B wheat plants. Unfortunately, this experiment requires at least one year (8 months for wheat transformation and additional months for crossing with different +B genotypes).

The complete inactivation of DCR28 represents a challenge because of the 15 members of the DCR28 gene family.

However, to find additional support for whether DCR28 is a drive-controlling gene, we asked whether also the B of the rye-related species *Aegilops speltoides* encodes a first pollen mitosis-active DCR28-like gene. *Ae. speltoides* was selected because of the rye-similar B chromosome drive mechanism (Wu et al. 2019). Our transcriptome analysis identified a first pollen mitosis active, B-specific DCR28-like gene in *Ae. speltoides*. The same gene B-specific gene was found in four different accessions of *Ae. speltoides*. Thus, B-specificity of DCR28-like in related species and first pollen mitosis activity supports the likely drive controlling function of DCR28

We added the following to the manuscript: “In addition, we tested whether the B chromosome of the related species *Aegilops speltoides* also encodes a first pollen mitosis-active DCR28-like gene. *Ae. speltoides* was selected because of the rye B chromosome-like drive mechanism (Wu et al. 2019). BLAST search of the *de novo* assembled transcriptome of anthers undergoing PMI from *Ae. speltoides* with B chromosomes identified a DCR28-like gene (called AesDCR288). AesDCR28 is 290-aa long and has 60% (161/265) positive amino acids compared to DCR28 of the rye B (Suppl. Fig. 9). Like in rye, the B paralog of DCR28 was shorter than the corresponding A chromosome variant. Further, genomic PCR showed the presence of AesDCR28 on all B chromosomes from four different *Ae. speltoides* accessions (Fig. 4c). Thus, both B chromosomes of rye and *Ae. speltoides* encode DCR28-like genes.”

Based in this findings we updated the phylogenetic tree of DCR28 with the recent *Ae. speltoides* data. Now it reads:

The phylogeny of *DCR28*

DCR28 is classified as a member of the microtubule-associated Futsch-like protein family (PTHR34468) by PANTHER. To better understand its evolution, a phylogenetic tree of *DCR28*-like genes was built based on 434 protein sequences of 325 species obtained from the NCBI database. Members of this gene family were detected throughout the major plant groups, from algae to angiosperms. Rooting the tree with *Chara brownii*, an alga representing the phylogenetically earliest branch in our dataset, we found that an early gene duplication resulted in two major groups within the *DCR28*-related genes (Fig. 7). This duplication must have happened at, or after the origin of the angiosperms, as in algae, sporophytes and gymnosperms we found only one type of the genes. However, as gymnosperms possess a copy that groups at the base of type 1 of the gene family (blue in Fig. 7) and, at least in some

analyses, the fern sequence of *Adiantum nelumboides* groups at the base of type-2 sequences (Supplementary Fig. 13) the duplication could have also happened within early seed plants. Not all angiosperm species in our dataset carry both types of *DCR28*-related proteins. It seems that type-2 sequences (red in Fig. 7) were particularly lost in many branches of angiosperms. The *DCR28* sequence groups with *Triticum*- and *Triticeae*-derived genes within the grass clade of type 2, placing its origin within this group.

5. Do homologous genes for these candidate genes also exist on the B chromosomes of other species?

RESPONSE: This is an excellent question. To answer this question, we generated the RNA-seq data of anthers undergoing the first pollen mitosis of *Ae. speltoides* with 3 B chromosomes. The drive mechanism of *Ae. speltoides* B chromosome is similar to the B chromosome of rye (Wu et al. 2019). Trinity was used to *de novo* assemble RNA-seq data of pollen undergoing the first mitosis into transcriptome. Strikingly, of the five rye B-drive candidate genes, only *DCR28* is present in the *de novo* assembled transcriptome. Genomic PCR showed that it was present on all B chromosomes from four different *Ae. speltoides* accessions (Fig. 4c). The 290-aa *DCR28* of *Ae. speltoides* B has 60% (161/265) positive amino acids compared to the 258-aa *DCR28* of rye B (Suppl. Fig. 9). Therefore, all B chromosomes of rye and *Ae. speltoides* encode *DCR28*.

6. The authors identified five candidate genes that control rye B chromosome drive. I would be interested to hear the authors' thoughts on whether B chromosome drive is controlled by a single gene or regulated by multiple genes. Are there other factor may involved in this regulation, and if these genes were transferred to other chromosomes, would they induce nondisjunction?

RESPONSE: The process of the B chromosome drive in rye is very likely not controlled by a single gene. The segregation failure of the B chromatids at pollen first pollen mitosis probably involves a complex interplay of multiple genes/proteins, particularly *trans*-acting drive controlling gene(s) located at the terminal end of the long B chromosome arm (*DCR*) and the *cis*-active critical site for nondisjunction, at the B chromosome pericentromere. The pioneering work from (Endo et al. 2008) revealed that the wheat-rye B translocated chromosome B^s-10, which carries the drive control but contains a wheat pericentromere, behaves like wheat chromosomes and lost the drive.

Minor comments:

In Fig. 4, bold the text in (c).

RESPONSE: Many thanks, and we took this suggestion.

Line 78: The ~560 Mb... (instead of Mbp)

RESPONSE: Many thanks, and we took this suggestion.

Reviewer #3 (Remarks to the Author):

General comment

The manuscript entitled “The genetic mechanism of B chromosome drive in rye illuminated by chromosome-scale assembly”, by Chen et al., addresses the main question about B chromosomes in one of the best known systems, that of rye, reaching hitherto almost unknown depth about the molecular mechanisms by which these genomic parasites manage to establish and maintain themselves in populations. The presence of functional genes for ribosomal DNA was long ago inferred from nucleolus formation by grasshopper B chromosomes (Bidau 1986; López-León et al. 1991). The first protein genes in B chromosomes were reported in the migratory locust, but they were also for repetitive DNA, specifically coding for the H3 and H4 histone genes, and their possible functionality was unknown (Teruel et al. 2010). However, the presence of functional protein genes on B chromosomes was ignored for long, simply because their search was technically unapproachable for chromosomes whose DNA is mostly repetitive. Now, thanks to recent improvements in massive sequencing techniques, yielding longer and longer reads, the authors of this manuscript have reached a major milestone in assembling a ~430 Mb large pseudomolecule representing ~77% of the rye B. Until now, the closest thing to this was a 297 Mb assembly in 444 scaffolds for the B chromosome of *Nasonia vitripennis* (Benetta et al. 2020), and 125.9 Mb of for the B chromosome of maize, representing 89% of its predicted 141-Mb size (Blavet et al. 2021). Therefore, the work carried out in this manuscript is very meritorious, as rye genome is thrice that of maize, and 23 times larger than that of *Nasonia*. In fact, it will be very difficult to perform a work like this in non-model organisms lacking the technical facilities existing in rye.

But even more important is the finding of a couple of genes functionally involved in B drive. In *Nasonia*, a gene involved in drive (named haploidizer) was also found, but this was of bacterial origin and was not found in the A genome of the species. In maize, no functional genes could be ascribed as responsible for B drive (Blavet et al. 2021). In rye, however, the two genes found (DCR28 and DCR400) are also part of the A genome of rye and, at last, we are witnessing how the drive of a B chromosome is possible thanks to the involvement of genes derived from the host genome, an unprecedented knowledge until now, thanks to the present manuscript.

The methodological approach has been the basis of the success of the achievements of this research. Having the possibility of sequencing the B chromosome in the wheat genome, instead of the host (rye) genome, has avoided interference from sequence homology between the B and A genes of rye. Also carrying out the functional analysis of the candidate genes responsible for B drive is an additional merit which logically increases the quality of this manuscript even more. But I want to highlight that this work has been possible thanks to the previous work by some Newton's giants (e.g. Neil Jones in UK and María Puertas in Spain), who laid the foundations over many years to make it possible to get here, and who, in my opinion, should be given more credit in this historical paper on rye B chromosomes.

RESPONSE: We agree, and much appreciate the previous research on the rye B chromosome by Maria Puertas and Neil Jones. Additional publications authored by both colleagues were added to the manuscript.

Of course, it had to be Andreas Houben's group that achieved this milestone, which rounds off his career of technical excellence since I met him for the first time, in 1993, during the first B chromosome conference, where he presented the chromosome microdissection technique, which many of us have done later when he was already immersed in other innovations. That said, I have no doubt that this work should be published and, with the aim of contributing as much as possible to improving the presentation, so that a work as complex and dense as this one is accessible to a general audience, I make the following minor comments.

RESPONSE: Many thanks for the very kind comments.

Minor

comments:

Lines 53-61. I think that this paragraph explaining the evolutionary arms race between B chromosome and their host genome should be revised, as, to my understanding, some of the cause-consequence descriptions are presented backwards. Parasites in general are not

considered parasites because they cause abnormal phenotypes, stress, disease or reduced fertility. Rather, they are considered parasites because they exploit host mechanisms to ensure their own transmission and maintenance within populations. Stress, disease or reduced fertility are byproducts (side-effects) of parasitism. Likewise, B chromosomes are considered parasitic because they have mechanisms (chromosome drive, in this case) that allows them to use the cellular machinery to achieve their own transmission, even if they are neutral or detrimental to the host. Drive is also not only a mechanism to by-pass suppressors. If the chromosome was not driving before the appearance of suppressors, suppressors would not arise, but the chromosome would be eliminated by drift in a few generations. Therefore, the evolutionary arms race model should begin with chromosome drive, then selection favouring the evolution of suppression, then stronger or different drive mechanisms, then stronger or different suppressors, etc. Arms race description in these lines highly fits the near-neutral model devised by Camacho et al. (1997, 2000) and Camacho (2005). González-Sánchez et al. (2003) applied this model to the case of maize. In my opinion, the near-neutral model also applies to B chromosomes in rye as the A-B chromosomes arms race explains why B-drive and A suppressor genes could have had successive turnovers across evolution, according to authors' discussion, and new genes could have been recruited to the B chromosome showing the "regeneration" stage of the model by which one B variant is replaced by another displaying a more efficient drive mechanism. As this manuscript demonstrates, this may be based on a new B-gene providing more efficient drive in new environmental circumstances, or else on a newly recruited gene coming from the A genome which provides this same advantage for the B in terms of its drive.

RESPONSE:

As suggested, we have rewritten this part of the introduction. In a future manuscript, we will elaborate more on the possible arms race model applicable to the B chromosome of rye. Now it reads: "B chromosomes are considered parasitic because they have mechanisms that allow them to use the cellular machinery of the host to achieve their own transmission, even if they are neutral or detrimental to the host. To avoid elimination, many B chromosomes exhibit a type of non-Mendelian inheritance termed *chromosome drive*, whereby segregation during cell divisions before, during, or after meiosis is biased in favour of increasing the number of B chromosomes in germinal cells (Chen et al. 2022). The long-term evolution of B chromosomes is likely the outcome of selection on the host genome to eliminate B chromosomes or suppress their effects and on the ability of the B chromosomes to escape through the generation of new variants (Camacho et al. 2000). This highly dynamic mode of evolution is consistent with the extreme variation in B chromosome numbers between species, individuals, and tissues within individuals."

Line 65: "Drive mechanisms in B chromosome systems have been studied in many species"
How many, approximately? Can the authors provide some proportions and references?

RESPONSE: To keep our manuscript as concise as possible and focus on the drive of the rye B, we did not list publications where the drive of B chromosomes was studied. Instead we referred to two excellent reviews on this topic ("for reviews: (Camacho 2022; Jones 2018)"). Unfortunately, despite many B drive papers, only one publication exists where a B drive-controlling gene was identified and tested (Dalla Benetta et al. 2020). We added this information. Now it reads: "The only known link between a B chromosome encoded gene and chromosomal drive is the gene haplodizer in the parasitoid wasp *Nasonia vitripennis* (Dalla Benetta et al. 2020)".

Lines 137: "A previously-undocumented deficient B variant named Bk-3 was identified when we screened the progeny of wheat with Bk-2" Do the authors mean that this variant arose de novo among the progeny of wheat with Bk-2?

RESPONSE: Yes, B^k-3 is likely a spontaneous de novo product of chromosome B^k-2. We found the complete loss of E3900-signals when we screened the progenies of plant with B^k-2.

Figure 3c and 3d: It would be helpful to include the approximate location of the drive control region in Figure 3c as well, for clarity.

RESPONSE: Many thanks and we took this suggestion.

Lines 246, Suppl. Table 4: Header of column A is missing. Do asterisks mean something in column H? Information in columns I and J (primary and secondary confidence class) are not explained in the manuscript. Also, to facilitate reading, it would be better that each supplemental file contains its table caption.

RESPONSE: Many thanks for finding the problems and the suggestion of adding table captions to each supplemental file. We removed all the asterisks in column H as they have no special meaning (Suppl. Table 4). About columns I and J (primary and secondary confidence class), they are the confidence classes from the published rye genome annotation. The reviewer is correct in that it has no meaning for B-located copies of genes. We apologies for keeping the information in the table. We remove the two column from the table.

Line 259. Figure 4a is a bit hard to understand, please consider redesigning, or including more explanations in the legend.

RESPONSE: We redesigned figure 4a and extended the legend. Now it reads: **(a)** Comparative RNA-seq analysis to identify candidates in charge of the drive. The data included 18 RNA-seq sets from wheat with different numbers of standard Bs (0B, 2B^s, and 4B^s) and of 0B and 2B rye (Supplementary Table 3). 15 RNA-seq data derived from wheat B variants with (1B^k-2) or without the ability to drive (1B^s-8, 1B^k-1) and from rye carrying a drive negative deficient B (1defB). Next, we performed differential expression analysis between plants with and without B drive (fold change > 2, p-adjusted value < 0.01) in wheat and rye to find commonly drive up-regulated genes. First, we pair-wise compared the drive-positive genes of wheat carrying the different B numbers and variants (2B^s, 4B^s, and 1B^k-2) with 0B, respectively. 533 commonly up-regulated differentially expressed genes (DEGs) were identified in three comparisons. Next, the genes of wheat with 2B^s, 4B^s, and 1B^k-2 were compared to the genes of the drive-negative wheat+ 1B^s-8, and 305 up-regulated DEGs remained. Afterwards, the genes of wheat with 2B^s, 4B^s, and 1B^k-2 were compared with the data of the drive-negative wheat+1B^k-1, and 29 commonly up-regulated DEGs were found. Finally, after comparing the data of rye with 2B to the data of 0B rye and rye+1defB, respectively, a total of 23 commonly up-regulated DEGs were found across the 11 comparisons. CD-HIT clustering revealed that the 23 DEGs belong to 7 single-copy genes and 2 gene families with multiple members (Supplementary Table 6). The left side indicates the 4 drive-positive data sets (horizontal) that were compared with the 5 drive-negative data sets (lateral), resulting in 11 comparisons between drive-positive data and drive-negative data (black circles). '∩' indicates intersection among comparisons. On the right side the number of differentially expressing candidate genes are indicated for the line-wise comparison shown on the left. The color code reflects the progressing reduction of candidate genes with each analysis step.

Lines 271: “Four genes (DCR398, DCR399, DCR145, and DCR154) appear to be located not only within the DCR but also at other positions on the rye B chromosome” Since this conclusion is based on PCR results, and there is no 0B control, I wonder if it is possible that these PCR amplifications might be due to primers inespecifically targeting homologous sequences from genes located somewhere in the A chromosomes of the wheat genome. However, in the

methods section they mention “B chromosome-specific primers”, which may indicate that these primers were designed to amplify only the B chromosome version. Please, clarify this point.

RESPONSE: Many thanks for this suggestion. To confirm the B chromosome-specificity of the applied primer pairs, we performed PCR using +B wheat and 0B wheat DNA as a template. Except for DCR154 (as it is a widely distributed transposase) we confirmed the B-specificity of applied DCR398, DCR399 and DCR154-specific primers. This test result has been added to Supplementary Fig.6a.

Lines 294: “The Futsch-like family is unknown in plants” Do the authors mean the function is not described in plants? In a quick database search in Uniprot, I found several entries of “microtubule associated futsch-like protein” in *Arabidopsis thaliana*, for instance, so the protein is known in plants; although papers describing its function are all in animals.

RESPONSE: To our knowledge, the Futsch-like family has not yet been explored in plants. The descriptor "microtubule associated futsch-like protein" is based only on in silico sequencing data comparisons. To make it more precise now it reads: “The function of the Futsch-like family is unknown in plants, ...”.

Line 300: "DCR400 not only shows stronger expression in shoot apical meristems" Figure 6a shows differential expression of DCR28 and DCR400 in both A and B chromosomes. Perhaps I have not been able to find in the manuscript how A and B derived transcripts were identified. In case of DCR28, perhaps transcript length worked but, in DCR400, lines 352-354 indicate that “DCR400 shows exceptionally high amino acid identity to the rye paralog gene”. In case of B-carrying rye plants, both A - and B-derived transcripts need to be identified to build a Figure like 6a. If it is not described in the manuscript, please do so. In case of DCR400, Figure 6a shows that DCR400 expression was much higher in 2B than in 1defB and 0B, but the TPM is roughly 50/50 distributed between DCR400 and DCR400-like rye. I wonder if this might be caused by the mapping algorithm not being able to distinguish between DCR400 and DCR400-like rye, and thus it mapped roughly half the reads to each of them in most of their length. Even so, the amount of DCR400 transcripts was higher in 2B than 0B and 1defB suggesting the differential expression of this gene in 2B plants. Please clarify this point.

RESPONSE: Many thanks. To answer the question of DCR400, we aligned the RNA sequence of B-copy to the A-copy; their identities are 2577/2620(98%) on RNA level. Even if the B-copy and A-copy share high amino acid identity, they still have enough single nucleotide

polymorphisms (> 1 SNP/100 bp) to let the bioinformatic tool differentiate the RNA reads from them.

Line 336: "Local gene clusters are suggestive of duplication events mediated by local microhomologies via (e.g.) repeat arrays that lead to strand slippage during replication" There has been several instances of meiotic drivers that have been amplified via tandem duplication (eg. Slx11 and Sly in mouse; Cocquet et al. 2009; Kruger et al. 2019) or somehow spread in association with other repetitive DNA or transposable sequences, such as wtf spore killers in *S. pombe* (Eickbush et al. 2019) and Dox in *Drosophila* (Muirhead et al. 2021). Yet the mechanism by which they amplified is still not demonstrated. I suggest to cite these papers here, making clear that the slippage is hypothetical, also discussing some alternative hypotheses, if applicable (eg. ectopic recombination, nonallelic gene conversion...).

RESPONSE: We agreed and added the following: "Alternatively, processes like tandem duplication (Kruger et al. 2019) or spreading in association with other repetitive DNA or transposable sequences (Eickbush et al. 2019), or others played a role".

Lines 350-351: "though DCR28 encodes a protein 71 residues shorter than its A chromosome counterpart" Maybe I missed it, but it is not clear to me how the rye B transcripts are assembled, neither how the author assign them to the B chromosome. Please provide more details on this, and also make clear that this is a predicted protein based on nucleotide sequence from those transcripts.

RESPONSE: Many thanks for the question. We assembled the genomic and transcriptomic sequences in the background of wheat plus rye B chromosomes. The lower sequence similarity between the wheat A chromosomes and the rye B chromosome sequences than rye A and B chromosomes allowed the identification of rye B sequences and transcripts. This "trick" was essential for the identification of B-encoded transcripts and genes.

We first generated the reference genomic sequences of the rye B chromosome based on long-read sequencing. After we mapped the RNA-seq data to the reference genome of wheat plus rye B chromosome and the alignment was processed to produce a gene feature annotation with StringTie (v 2.1.1, default parameters). A set of non-redundant transcripts was generated

using gffread (Version v0.12.6). So the rye B transcripts are assembled. TransDecoder (v 5.5.0) was used to annotate coding regions within transcripts.

Line 354: "suggesting a more recent copying event" Or, alternatively, the existence of high functional constrains for B drive.

RESPONSE: We agree and added this option. Now it reads: *DCR400* shows exceptionally high amino acid identity to the rye paralog gene SECCE1Rv1G0060580 (called *DCR400-like rye*) of A chromosome 1R (Supplementary Fig. 11e) suggesting a more recent copying event or the existence of high functional constrains for B chromosome drive.

Line 517: Change "has also be reported" for "has also been reported"

RESPONSE: Has been corrected.

Lines 577-578: "To counteract purifying selection, the gain of a chromosome drive mechanism is a prerequisite for the transgenerational stability of a newly formed B" As beneficial effects of the newly formed proto-B are unlikely, the safest pathway by which it can remain in a population is by displaying drive from the beginning. This initial drive does not necessarily has to be the same observed for current rye Bs. The existence of many gene copies of the driver gene, with some sequence differences between them, would support that drive mechanism could have changed across evolution, probably due to an arms race dynamics. In the beginning, drive could even occur prior or during meiosis, and later the appearance of other drive mechanisms might mask the ancestral mechanisms. These comments complement those about Lines 53-61, as the extraordinary molecular details revealed in the present manuscript indicate that the B chromosome system in rye clearly fits the near-neutral model of B chromosome evolution.

RESPONSE: We agree and added the following: "This initial drive does not necessarily have to be the same observed for current rye Bs. The existence of many gene copies of the driver candidate DCR28, with some sequence differences between them, would support that drive mechanism could have changed across evolution, probably due to an arms race dynamics".

References

- Bidau, C. J. (1986). A nucleolar-organizing B chromosome showing segregation–distortion in the grasshopper *Dichroplus pratensis* (Melanoplinae, Acrididae) . *Canadian Journal of Genetics and Cytology*, 28(1), 138–148. <https://doi.org/10.1139/g86-018>
- Benetta, E. D., Antoshechkin, I., Yang, T., Nguyen, H. Q. M., Ferree, P. M., & Akbari, O. S. (2020). Genome elimination mediated by gene expression from a selfish chromosome. *Science Advances*, 6(14), eaaz9808. <https://doi.org/10.1126/sciadv.aaz9808>
- Blavet, N., Yang, H., Su, H., Solanský, P., Douglas, R. N., Karafiátová, M., Šimková, L., Zhang, J., Liu, Y., Hou, J., Shi, X., Chen, C., El-Walid, M., McCaw, M. E., Albert, P. S., Gao, Z., Zhao, C., Ben-Zvi, G., Glick, L., ... Bartoš, J. (2021). Sequence of the supernumerary B chromosome of maize provides insight into its drive mechanism and evolution. *Proceedings of the National Academy of Sciences of the United States of America*, 118(23), 1–11. <https://doi.org/10.1073/PNAS.2104254118>
- Camacho, J. P. M., Shaw, M. W., López-León, M. D., Pardo, M. C., & Cabrero, J. (1997). Population dynamics of a selfish B chromosome neutralized by the standard genome in the grasshopper *Eyprepocnemis plorans*. *The American Naturalist*, 149(6), 1030-1050.
- Camacho, J. P. M., Sharbel, T. F., & Beukeboom, L. W. (2000). B-chromosome evolution. *Philosophical Transactions of the Royal Society of London. Series B: Biological Sciences*, 355(1394), 163-178.
- Camacho, J. P. M. (2005). B chromosomes. In *The evolution of the genome* (pp. 223-286). Academic Press.
- Cocquet, J., Ellis, P. J., Yamauchi, Y., Mahadevaiah, S. K., Affara, N. A., Ward, M. A., & Burgoyne, P. S. (2009). The multicopy gene *Sly* represses the sex chromosomes in the male mouse germline after meiosis. *PLoS biology*, 7(11), e1000244.
- Eickbush, M. T., Young, J. M., & Zanders, S. E. (2019). Killer meiotic drive and dynamic evolution of the *wtf* gene family. *Molecular Biology and Evolution*, 36(6), 1201-1214.
- González-Sánchez, M., González-González, E., Molina, F., Chiavarino, A. M., Rosato, M., & Puertas, M. J. (2003). One gene determines maize B chromosome accumulation by preferential fertilisation; another gene (s) determines their meiotic loss. *Heredity*, 90(2), 122-129.
- Kruger, A. N., Brogley, M. A., Huizinga, J. L., Kidd, J. M., de Rooij, D. G., Hu, Y. C., & Mueller, J. L. (2019). A neofunctionalized X-linked ampliconic gene family is essential for male fertility and equal sex ratio in mice. *Current Biology*, 29(21), 3699-3706.
- López-León, M., Cabrero, J., & Camacho, J. (1991). A nucleolus organizer region in a B chromosome inactivated by DNA methylation. *Chromosoma*, 100(2), 134–138.

<http://www.springerlink.com/index/U10420564J560M64.pdf>

Muirhead, C. A., & Presgraves, D. C. (2021). Satellite DNA-mediated diversification of a sex-ratio meiotic drive gene family in *Drosophila*. *Nature Ecology & Evolution*, 5(12), 1604-1612.

Teruel, M., Cabrero, J., Perfectti, F., & Camacho, J. P. M. (2010). B chromosome ancestry revealed by histone genes in the migratory locust. *Chromosoma*, 119(2), 217–225.

<https://doi.org/10.1007/s00412-009-0251-3>

Reviewer #4 (Remarks to the Author):

I co-reviewed this manuscript with one of the reviewers who provided the listed reports. This is part of the Nature Communications initiative to facilitate training in peer review and to provide appropriate recognition for Early Career Researchers who co-review manuscripts

RESPONSE: *Many thanks for your help and time.*

REFERENCES

Banaei-Moghaddam AM, Schubert V, Kumke K, Weibeta O, Klemme S, Nagaki K, Macas J, Gonzalez-Sanchez M, Heredia V, Gomez-Revilla D, Gonzalez-Garcia M, Vega JM, Puertas MJ, Houben A (2012) Nondisjunction in favor of a chromosome: the mechanism of rye B chromosome drive during pollen mitosis. *Plant Cell* 24 (10):4124-4134. doi:10.1105/tpc.112.105270

Blavet N, Yang H, Su H, Solanský P, Douglas RN, Karafiátová M, Šimková L, Zhang J, Liu Y, Hou J (2021) Sequence of the supernumerary B chromosome of maize provides insight into its drive mechanism and evolution. *Proceedings of the National Academy of Sciences* 118 (23):e2104254118

Camacho JPM (2022) Non-Mendelian segregation and transmission drive of B chromosomes. *Chromosome Res* 30 (2-3):217-228. doi:10.1007/s10577-022-09692-7

Camacho JPM, Sharbel TF, Beukeboom LW (2000) B-chromosome evolution. *Philosophical Transactions of the Royal Society of London Series B-Biological Sciences* 355 (1394):163-178

Chen J, Birchler JA, Houben A (2022) The non-Mendelian behavior of plant B chromosomes. *Chromosome Res.* doi:10.1007/s10577-022-09687-4

- Chen J, Tang Y, Yao L, Wu H, Tu X, Zhuang L, Qi Z (2019) Cytological and molecular characterization of *Thinopyrum bessarabicum* chromosomes and structural rearrangements introgressed in wheat. *Molecular Breeding* 39 (10):1-14
- Dalla Benetta E, Antoshechkin I, Yang T, Nguyen HQM, Ferree PM, Akbari OS (2020) Genome elimination mediated by gene expression from a selfish chromosome. *Sci Adv* 6 (14):eaaz9808. doi:10.1126/sciadv.aaz9808
- De Carvalho M, Jia G-S, Srinivasa AN, Billmyre RB, Xu Y-H, Lange JJ, Sabbarini IM, Du L-L, Zanders SE (2022) The wtf meiotic driver gene family has unexpectedly persisted for over 100 million years. *Elife* 11:e81149
- Eickbush MT, Young JM, Zanders SE (2019) Killer meiotic drive and dynamic evolution of the wtf gene family. *Mol Biol Evol* 36 (6):1201-1214. doi:10.1093/molbev/msz052
- Endo TR, Nasuda S, Jones N, Dou Q, Akahori A, Wakimoto M, Tanaka H, Niwa K, Tsujimoto H (2008) Dissection of rye B chromosomes, and nondisjunction properties of the dissected segments in a common wheat background. *Genes & genetic systems* 83 (1):23-30
- Jones RN (2018) Transmission and drive involving parasitic B chromosomes. *Genes-Basel* 9 (8). doi:ARTN 388
10.3390/genes9080388
- Kruger AN, Brogley MA, Huizinga JL, Kidd JM, de Rooij DG, Hu YC, Mueller JL (2019) A neofunctionalized X-linked ampliconic gene family is essential for male fertility and equal sex ratio in mice. *Current Biology* 29 (21):3699-+. doi:10.1016/j.cub.2019.08.057
- Langdon T, Seago C, Jones RN, Ougham H, Thomas H, Forster JW, Jenkins G (2000) De novo evolution of satellite DNA on the rye B chromosome. *Genetics* 154 (2):869-884
- Martis MM, Klemme S, Banaei-Moghaddam AM, Blattner FR, Macas J, Schmutzer T, Scholz U, Gundlach H, Wicker T, Šimková H (2012) Selfish supernumerary chromosome reveals its origin as a mosaic of host genome and organellar sequences. *Proceedings of the National Academy of Sciences* 109 (33):13343-13346
- Mendelson D, Zohary D (1972) Behavior and transmission of supernumerary chromosomes in *Aegilops speltoides*. *Heredity* 29 (Dec):329-&
- Ou S, Chen J, Jiang N (2018) Assessing genome assembly quality using the LTR Assembly Index (LAI). *Nucleic acids research* 46 (21):e126-e126
- Rhie A, Nurk S, Cechova M, Hoyt SJ, Taylor DJ, Altemose N, Hook PW, Koren S, Rautiainen M, Alexandrov IA (2023) The complete sequence of a human Y chromosome. *Nature*:1-11
- Schlebusch SA, Rídl J, Poinet M, Ruiz-Ruano FJ, Reif J, Pajer P, Pačes J, Albrecht T, Suh A, Reifová R (2023) Rapid gene content turnover on the germline-restricted chromosome in songbirds. *Nature Communications* 14 (1):4579

Wu D, Ruban A, Fuchs J, Macas J, Novak P, Vaio M, Zhou Y, Houben A (2019) Nondisjunction and unequal spindle organization accompany the drive of *Aegilops speltoides* B chromosomes. *New Phytol* 223 (3):1340-1352. doi:10.1111/nph.15875

REVIEWER COMMENTS

Reviewer #1 (Remarks to the Author):

The presentation is much improved. The evidence that the authors have identified a trans-acting factor for the rye B is reasonable. Some suggestions on the current version follow.

Line 80, need a space between “although” and “a”

RESPONSE: Many thanks and we have corrected this mistake.

Line 82, best to delete the word “possibly”. The cis site for the maize B was well defined years ago (Han et al 2007, Plant Cell 19: 524-533) and a trans-acting factor even earlier (Birchler and Auger, 2002, J. Hered. 93: 42-47). The cited paper is fine because therein they were placed on the currently available maize B sequence.

RESPONSE: Many thanks and we took this suggestion.

Line 97, Delete “(germline)”

RESPONSE: Many thanks and we took this suggestion.

Line 107, Insert “the” between “at’ and “first”

RESPONSE: Many thanks and we have corrected this mistake.

Line 109, Insert “a” between “of” and “standard”

RESPONSE: Many thanks and we have corrected this mistake.

Line 243, Remove extra space

RESPONSE: Many thanks and we have corrected this mistake.

Line 315, Delete “s” from “plays”

RESPONSE: Many thanks and we have corrected this mistake.

Line 415, Delete extra “that”

RESPONSE: Many thanks and we have corrected this mistake.

Line 419, Delete “of the genes”

RESPONSE: Many thanks and we have corrected this mistake.

Line 484, Insert “the” between “of” and “rye”

RESPONSE: Many thanks and we have corrected this mistake.

Lines 590 to 623, The authors suggest a scenario that chromosomes are missegregated and land in micronuclei in gamete development and undergo chromoanagenesis, rearrange, and organize again as a B chromosome. But then later, the authors wind up claiming that all 7 chromosomes get involved somewhat sequentially. Looking at Supplemental Figure 17, it does seem that chromosomes 6 and 7 were early contributors. The position on the B chromosome is given but how does a synteny relationship look like between those 6 and 7 paralogs on the B versus 6 and 7 paralogs on the A chromosomes, respectively? If there is good synteny between the B and A paralogs on 6 and 7, then rearrangement seems unlikely. The claim is that more chromoanagenesis occurred for the other five chromosomes in subsequent events. But why would the pieces insert into the B chromosome specifically rather than being splattered all over the genome? Rye and barley have good synteny, so that seems unlikely. I guess one could argue that such fragments landing in the A chromosomes would be selected against due to aneuploidy, but the same would apply to the B chromosome unless the genes are silenced, which the RNA data suggests otherwise. One can see that as one builds on this scenario, it might be an “air castle”. One solution would be to delete this whole section and Suppl Figure 17. Further work might eventually lead to an understanding that seems reasonable.

RESPONSE: Since two of the referees did not favour our working hypothesis on the origin of the B chromosome and the central message of our manuscript is the the genetic mechanism of the rye B chromosome drive illuminated by chromosome-scale assembly, we decided to remove Suppl. Figure 17 and the corresponding text as suggested by you. We agree that it is more reasonable to develop our hypothesis of the B chromosome origin further based on additional experimental data in the future.

Reviewer #2 (Remarks to the Author):

After reading the revised version and response letter, I think the authors provide thoughtful replies and corresponding revisions to all of my comments. I have no further comments.

RESPONSE: Many thanks for your help in improving the manuscript.

Reviewer #3 (Remarks to the Author):

Authors have adequately addressed my concerns and suggestions. However, I still have a few remarks, the most important concerning the analysis requested by Reviewer 1 about sequence identity between rye A and B chromosome paralogs (Suppl. Figure 17). In my opinion, BLASTN might not be the best tool for this analysis, as it throws multiple hits for every single region of the B chromosome; and the parameters used for this search are likely too relaxed (Section 17 of the Methods) and could lead to inaccurate results. I suggest performing this analysis with more stringent BLAST search parameters and limiting the search to the best hit (or else explaining the rationale for the used parameters). However, a better approach would be to perform whole genome alignment of each A chromosome to the B pseudomolecule (masking repeats, and using LASTZ, for example), or extracting the exon and intron sequences from the A chromosomes annotation (with bedtools) and map them to the rye B pseudomolecule (using minimap2, GMAP or STAR for instance). Then, the authors should be able to

calculate a gene-based or window-based divergence between A and B genes.

Also, Suppl. Figure 17 contains mixed information about introns and exons, which is tricky because the models for molecular evolution of introns and exons are quite different. Most substitutions in introns are expected to be neutral, therefore the number of substitutions in introns (molecular divergence) is expected to be somewhat proportional to time. Substitutions in exons, however, may be constrained by selection. Additionally, introns may contain transposable elements (TEs) or other types of repeats, which could generate noise in inferring the A chromosome gene contribution to the B chromosome. For this task to be reliable, it is absolutely necessary to avoid the presence of repetitive DNA that may be present on two or more A chromosomes. Authors discarded genes annotated as TEs, but they did not mask repeats in the assemblies (at least I have not found information about this in the revised manuscript). For instance, the presence of other kinds of repeats, such as tandem repeats or TE fragments, inserted within the introns of the genic sequences, might give misleading results, since they could be present in several A chromosomes. Even masking known TEs, it is still conceivable the presence of a few still undescribed TEs in the rye genome which could not be masked. These repeats or repeat fragments could explain why all the length of the B chromosome pseudomolecule shows homology with most A chromosomes, as shown in Suppl. Figure 17. For this reason, I believe that restricting the analysis to exonic sequences would give a clearer picture of B chromosome origin. In any case, and given that I did not suggest this analysis, as far as I am concerned, it is not essential to maintain this new section and the authors could eliminate it from the manuscript if they consider it convenient.

RESPONSE: Since two of the referees did not favour our working hypothesis on the origin of the B chromosome and the central message of our manuscript is the the genetic mechanism of the rye B chromosome drive illuminated by chromosome-scale assembly, we decided to remove Suppl. Figure 17 and the corresponding text as suggested by you. We agree that it is more reasonable to develop our hypothesis of the B chromosome origin further based on additional experimental data in the future.

Minor comments.

- Authors provide the captions for Supplementary tables in the main text, but these captions are not included as headers within the tables themselves (xls files). In my opinion, the captions should be added to the xls files to facilitate reader comprehension. In the current version, the names of the Supplementary tables are not even shown in the file names, so readers must guess which of the nine supplementary tables they are looking at based on the information in the table.
- Figure S12b might include a typo in sequence names (NCR400 instead of DCR400).

RESPONSE: Many thanks and we have corrected this mistake.

- Figure S17 is presented in the Discussion rather than the Results section, which is fine. However, if it is kept in the final version, some information is missing. For subfigure A: Modify the X-axis label to clearly

indicate that it represents sequence identity with B-chromosomal sequences. The term "Frequency" typically refers to a proportion, but the color legend extends beyond 600. Please clarify whether this number represents the number of genes or something else (as per Section 17 of the Methods, I assume it might be the number of BLAST hits, which would indeed be higher). For subfigure B: please make it clear in the legend whether each dot represents a gene or a BLAST hit.

RESPONSE: Figure S17 has been removed as explained above.

Reviewer #4 (Remarks to the Author):

RESPONSE: Many thanks for your help in improving the manuscript.